# MEASURING THE RELIABILITY OF REINFORCEMENT LEARNING ALGORITHMS

**Stephanie C.Y. Chan,**[1]* **Samuel Fishman,**[1] **John Canny,**[1,2] **Anoop Korattikara,**[1] **& Sergio Guadarrama**[1]

[1]Google Research [2]Berkeley EECS

`{scychan,sfishman,canny,kbanoop,sguada}@google.com`

## ABSTRACT

Lack of reliability is a well-known issue for reinforcement learning (RL) algorithms. This problem has gained increasing attention in recent years, and efforts to improve it have grown substantially. To aid RL researchers and production users with the evaluation and improvement of reliability, we propose a set of metrics that quantitatively measure different aspects of reliability. In this work, we focus on variability and risk, both during training and after learning (on a fixed policy). We designed these metrics to be general-purpose, and we also designed complementary statistical tests to enable rigorous comparisons on these metrics. In this paper, we first describe the desired properties of the metrics and their design, the aspects of reliability that they measure, and their applicability to different scenarios. We then describe the statistical tests and make additional practical recommendations for reporting results. The metrics and accompanying statistical tools have been made available as an open-source library.[1] We apply our metrics to a set of common RL algorithms and environments, compare them, and analyze the results.

## 1 INTRODUCTION

Reinforcement learning (RL) algorithms, especially Deep RL algorithms, tend to be highly variable in performance and considerably sensitive to a range of different factors, including implementation details, hyper-parameters, choice of environments, and even random seeds (Henderson et al., 2017). This variability hinders reproducible research, and can be costly or even dangerous for real-world applications. Furthermore, it impedes scientific progress in the field when practitioners cannot reliably evaluate or predict the performance of any particular algorithm, compare different algorithms, or even compare different implementations of the same algorithm.

Recently, Henderson et al. (2017) has performed a detailed analysis of reliability for several policy gradient algorithms, while Duan et al. (2016) has benchmarked average performance of different continuous-control algorithms. In other related work, Colas et al. (2018) have provided a detailed analysis on power analyses for mean performance in RL, and Colas et al. (2019) provide a comprehensive primer on statistical testing for mean and median performance in RL.

In this work, we aim to devise a set of metrics that measure reliability of RL algorithms. Our analysis distinguishes between several typical modes to evaluate RL performance: "evaluation during training", which is computed over the course of training, vs. "evaluation after learning", which is evaluated on a fixed policy after it has been trained. These metrics are also designed to measure different aspects of reliability, e.g. reproducibility (variability *across* training runs and variability *across* rollouts of a fixed policy) or stability (variability *within* training runs). Additionally, the metrics capture multiple aspects of variability – dispersion (the width of a distribution), and risk (the heaviness and extremity of the lower tail of a distribution).

Standardized measures of reliability can benefit the field of RL by allowing RL practitioners to compare algorithms in a rigorous and consistent way. This in turn allows the field to measure

---

*Work done as part of the Google AI Residency

[1]https://github.com/google-research/rl-reliability-metrics

progress, and also informs the selection of algorithms for both research and production environments. By measuring various aspects of reliability, we can also identify particular strengths and weaknesses of algorithms, allowing users to pinpoint specific areas of improvement.

In this paper, in addition to describing these reliability metrics, we also present practical recommendations for statistical tests to compare metric results and how to report the results more generally. As examples, we apply these metrics to a set of algorithms and environments (discrete and continuous, off-policy and on-policy). We have released the code used in this paper as an open-source Python package to ease the adoption of these metrics and their complementary statistics.

| | | Dispersion (D) | Risk (R) |
|---|---|---|---|
| DURING TRAINING | **Across Time (T) (within training runs)** | IQR* within windows, after detrending | **Short-term:** CVaR$^\dagger$ on first-order differences 
 **Long-term**: CVaR$^\dagger$ on Drawdown |
| | **Across Runs (R)** | IQR* across training runs, after low-pass filtering. | CVaR$^\dagger$ across runs |
| AFTER LEARNING | **Across rollouts on a Fixed Policy (F)** | IQR* across rollouts for a fixed policy | CVaR$^\dagger$ across rollouts for a fixed policy |

Table 1: Summary of our proposed reliability metrics. For evaluation DURING TRAINING, which measures reliability over the course of training an algorithm, the inputs to the metrics are the performance curves of an algorithm, evaluated at regular intervals during a single training run (or on a set of training runs). For evaluation AFTER LEARNING, which measures reliability of an already-trained policy, the inputs to the metrics are the performance scores of a set of rollouts of that fixed policy. *IQR: inter-quartile range. $^\dagger$CVaR: conditional value at risk.

## 2 RELIABILITY METRICS

We target three different axes of variability, and two different measures of variability along each axis. We denote each of these by a letter, and each metric as a combination of an axis + a measure, e.g. "DR" for "Dispersion Across Runs". See Table 1 for a summary. Please see Appendix A for more detailed definitions of the terms used here.

### 2.1 AXES OF VARIABILITY

Our metrics target the following three axes of variability. The first two capture reliability "during training", while the last captures reliability of a fixed policy "after learning".

**During training: Across Time (T)** In the setting of evaluation during training, one desirable property for an RL algorithm is to be stable "across time" within each training run. In general, smooth monotonic improvement is preferable to noisy fluctuations around a positive trend, or unpredictable swings in performance.

This type of stability is important for several reasons. During learning, especially when deployed for real applications, it can be costly or even dangerous for an algorithm to have unpredictable levels of performance. Even in cases where bouts of poor performance do not directly cause harm, e.g. if training in simulation, high instability implies that algorithms have to be check-pointed and evaluated more frequently in order to catch the peak performance of the algorithm, which can be expensive. Furthermore, while training, it can be a waste of computational resources to train an unstable algorithm that tends to forget previously learned behaviors.

**During training: Across Runs (R)**   During training, RL algorithms should have easily and consistently reproducible performances across multiple training runs. Depending on the components that we allow to vary across training runs, this variability can encapsulate the algorithm's sensitivity to a variety of factors, such as: random seed and initialization of the optimization, random seed and initialization of the environment, implementation details, and hyper-parameter settings. Depending on the goals of the analysis, these factors can be held constant or allowed to vary, in order to disentangle the contribution of each factor to variability in training performance. High variability on any of these dimensions leads to unpredictable performance, and also requires a large search in order to find a model with good performance.

**After learning: Across rollouts of a fixed policy (F)**   When evaluating a fixed policy, a natural concern is the variability in performance across multiple rollouts of that fixed policy. Each rollout may be specified e.g. in terms of a number of actions, environment steps, or episodes. Generally, this metric measures sensitivity to both stochasticity from the environment and stochasticity from the training procedure (the optimization). Practitioners may sometimes wish to keep one or the other constant if it is important to disentangle the two factors (e.g. holding constant the random seed of the environment while allowing the random seed controlling optimization to vary across rollouts).

## 2.2   MEASURES OF VARIABILITY

For each axis of variability, we have two kinds of measures: dispersion and risk.

**Dispersion**   Dispersion is the width of the distribution. To measure dispersion, we use "robust statistics" such as the ***Inter-quartile range (IQR)*** (i.e. the difference between the 75th and 25th percentiles) and the ***Median absolute deviation from the median (MAD)***, which are more robust statistics and don't require assuming normality of the distributions. [2] We prefer to use IQR over MAD, because it is more appropriate for asymmetric distributions (Rousseeuw & Croux, 1993).

**Risk**   In many cases, we are concerned about the worst-case scenarios. Therefore, we define risk as the heaviness and extent of the lower tail of the distribution. This is complementary to measures of dispersion like IQR, which cuts off the tails of the distribution. To measure risk, we use the ***Conditional Value at Risk (CVaR)***, also known as "expected shortfall". CVaR measures the expected loss in the worst-case scenarios, defined by some quantile $\alpha$. It is computed as the expected value in the left-most tail of a distribution (Acerbi & Tasche, 2002). We use the following definition for the CVaR of a random variable $X$ for a given quantile $\alpha$:

$$\text{CVaR}_\alpha(X) = \mathbb{E}\left[X | X \leq VaR_\alpha(X)\right] \tag{1}$$

where $\alpha \in (0, 1)$ and the $VaR_\alpha$ (Value at Risk) is just the $\alpha$-quantile of the distribution of $X$. Originally developed in finance, CVaR has also seen recent adoption in Safe RL as an additional component of the objective function by applying it to the cumulative returns within an episode, e.g. Bäuerle & Ott (2011); Chow & Ghavamzadeh (2014); Tamar et al. (2015). In this work, we apply CVaR to the dimensions of reliability described in Section 2.1.

## 2.3   DESIDERATA

In designing our metrics and statistical tests, we required that they fulfill the following criteria:

- A minimal number of configuration parameters – to facilitate standardization as well as to minimize "researcher degrees of freedom" (where flexibility may allow users to tune settings to produce more favorable results, leading to an inflated rate of false positives) (Simmons et al., 2011).
- Robust statistics, when possible. Robust statistics are less sensitive to outliers and have more reliable performance for a wider range of distributions. Robust statistics are especially

---

[2]Note that our aim here is to measure the variability of the distribution, rather than to characterize the uncertainty in estimating a statistical parameter of that distribution. Therefore, confidence intervals and other similar methods are not suitable for the aim of measuring dispersion.

important when applied to training performance, which tends to be highly non-Gaussian, making metrics such as variance and standard deviation inappropriate. For example, training performance is often bi-modal, with a concentration of points near the starting level and another concentration at the level of asymptotic performance.

- Invariance to sampling frequency – results should not be biased by the frequency at which an algorithm was evaluated during training. See Section 2.5 for further discussion.

- Enable meaningful statistical comparisons on the metrics, while making minimal assumptions about the distribution of the results. We thus designed statistical procedures that are non-parametric (Section 4).

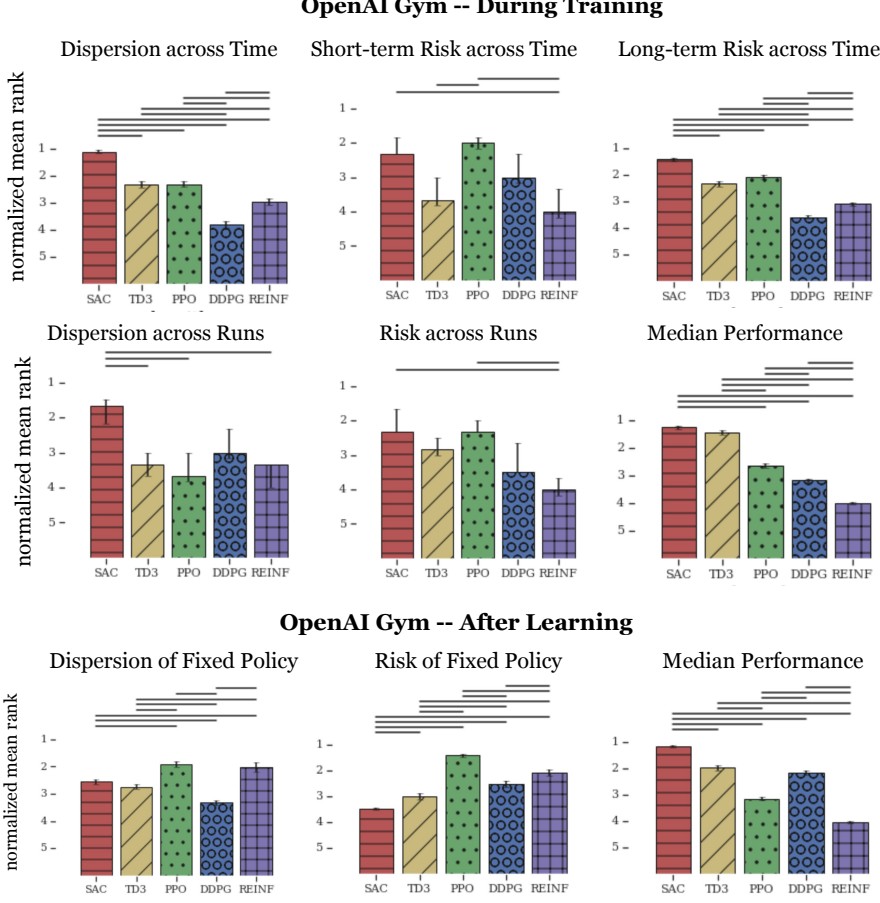

Figure 1: Reliability metrics and median performance for continuous control RL algorithms (DDPG, TD3, SAC, REINFORCE, and PPO) tested on OpenAI Gym environments. Rank 1 always indicates "best" reliability, e.g. lowest IQR across runs. Error bars are 95% bootstrap confidence intervals (# bootstraps = 1,000). Significant pairwise differences in ranking between pairs of algorithms are indicated by black horizontal lines above the colored bars. ($\alpha = 0.05$ with Benjamini-Yekutieli correction, permutation test with # permutations = 1,000). Note that the best algorithms by median performance are not always the best algorithms on reliability.

## 2.4 METRIC DEFINITIONS

**Dispersion across Time (DT): IQR across Time**   To measure dispersion across time (DT), we wished to isolate higher-frequency variability, rather than capturing longer-term trends. We did not want our metrics to be influenced by positive trends of improvement during training, which are in fact desirable sources of variation in the training performance. Therefore, we apply detrending before

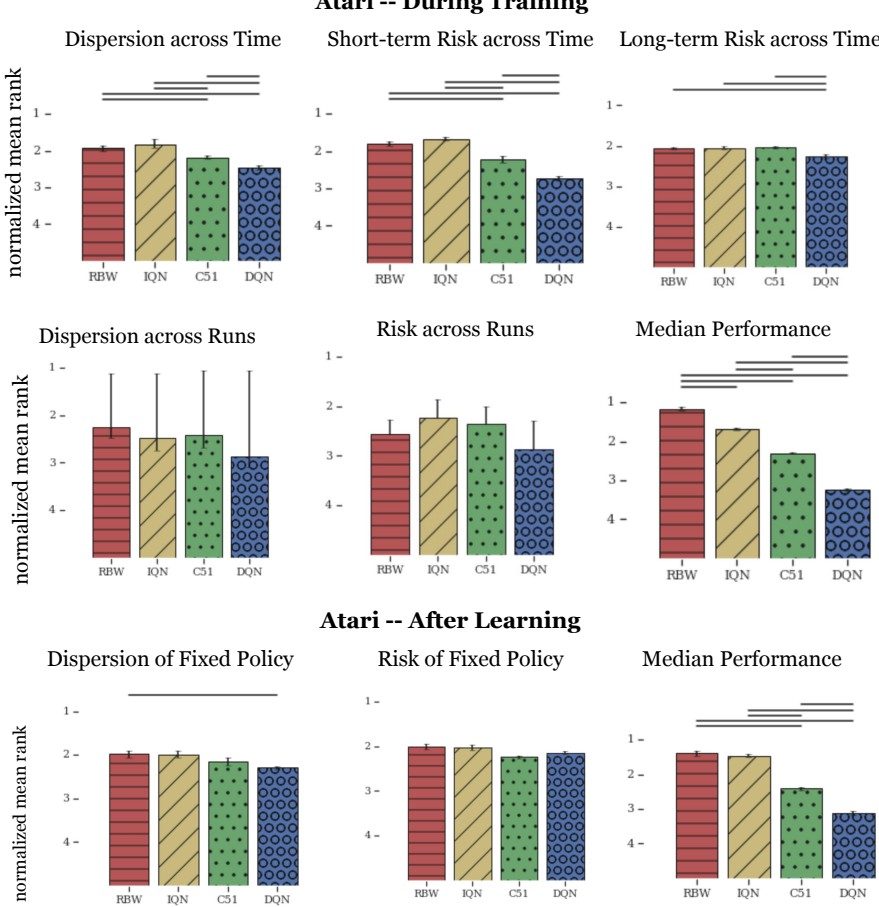

Figure 2: Reliability metrics and median performance for four DQN-variants (C51, DQN: Deep Q-network, IQ: Implicit Quantiles, and RBW: Rainbow) tested on 60 Atari games. Rank 1 always indicates "best" reliability, e.g. lowest IQR across runs. Significant pairwise differences in ranking between pairs of algorithms are indicated by black lines above the colored circles. ($\alpha = 0.05$ with Benjamini-Yekutieli correction, permutation test with # permutations = 1,000). Note that the best algorithms by median performance are not always the best algorithms on reliability. Error bars are 95% bootstrap confidence intervals (# bootstraps = 1,000).

computing dispersion metrics. For detrending, we used differencing (i.e. $y_t' = y_t - y_{t-1}$).[3] The final measure consisted of inter-quartile range (IQR) within a sliding window along the detrended training curve.

**Short-term Risk across Time (SRT): CVaR on Differences**   For this measure, we wish to measure the most extreme short-term drop over time. To do this, we apply CVaR to the changes in performance from one evaluation point to the next. I.e., in Eq. 1, $X$ represents the differences from one evaluation time-point to the next. We first compute the time-point to time-point differences on each training run. These differences are normalized by the distance between time-points, to ensure invariance to evaluation frequency (see Section 2.5). Then, we obtain the distribution of these differences, and find the $\alpha$-quantile. Finally, we compute the expected value of the distribution below the $\alpha$-quantile. This gives us the worst-case expected drop in performance during training, from one point of evaluation to the next.

---

[3]Please see Appendix B for a more detailed discussion of different types of detrending, and the rationale for choosing differencing here.

**Long-term Risk across Time (LRT): CVaR on Drawdown**   For this measure, we would also like to be able to capture whether an algorithm has the potential to lose a lot of performance relative to its peak, even if on a longer timescale, e.g. over an accumulation of small drops. For this measure, we apply CVaR to the ***Drawdown***. The Drawdown at time $T$ is the drop in performance relative to the highest peak so far, and is another measure borrowed from economics (Chekhlov et al., 2005). I.e. $\text{Drawdown}_T = R_T - \max_{t <= T} R_t$. Like the SRT metric, the LRT can capture unusually large short-term drops in performance, but can also capture unusually large drops that occur over longer timescales.

**Dispersion across Runs (DR): IQR across Runs**   Unlike the rest of the metrics described here, the dispersion across training runs has previously been used to characterize performance (e.g. Duan et al. (2016); Islam et al. (2017); Bellemare et al. (2017); Fortunato et al. (2017); Nagarajan et al. (2018)). This is usually measured by taking the variance or standard deviation across training runs at a set of evaluation points. We build on the existing practice by recommending first performing low-pass filtering of the training data, to filter out high-frequency variability within runs (this is instead measured using Dispersion across Time, DT). We also replace variance or standard deviation with robust statistics like IQR.

**Risk across Runs (RR): CVaR across Runs**   In order to measure Risk across Runs (RR), we apply CVaR to the final performance of all the training runs. This gives a measure of the expected performance of the worst runs.

**Dispersion across Fixed-Policy Rollouts (DF): IQR across Rollouts**   When evaluating a fixed policy, we are interested in variability in performance when the same policy is rolled out multiple times. To compute this metric, we simply compute the IQR on the performance of the rollouts.

**Risk across Fixed-Policy Rollouts (RF): CVaR across Rollouts**   This metric is similar to DF, except that we apply CVaR on the rollout performances.

## 2.5 INVARIANCE TO FREQUENCY OF EVALUATION

Different experiments and different tasks may produce evaluations at different frequencies during training. Therefore, the reliability metrics should be unbiased by the choice of evaluation frequency. As long as there are no cyclical patterns in performance, the frequency of evaluation will not bias any of the metrics except Long-Term Risk across Time (LRT). For all other metrics, changes in the frequency of evaluation will simply lead to more or less noisy estimates of these metrics. For LRT, comparisons should only be made if the frequency of evaluation is held constant across experiments.

## 3   RECOMMENDATIONS FOR REPORTING METRICS AND PARAMETERS

Whether evaluating an algorithm for practical use or for research, we recommend evaluating all of the reliability metrics described above. Each metric measures a different aspect of reliability, and can help pinpoint specific strengths and weaknesses of the algorithm. Evaluating the metrics is easy with the open-source Python package that we have released.

**Reporting parameters.**   Even given our purposeful efforts to minimize the number of parameters in the reliability metrics, a few remain to be specified by the user that can affect the results, namely: window size (for Dispersion across Time), frequency threshold for low-pass and high-pass filtering (Dispersion across Time, Dispersion across Runs), evaluation frequency (only for Long-term Risk across Time), and length of training runs. Therefore, when reporting these metrics, these parameters need to be clearly specified, and must also be held constant across experiments for meaningful comparisons. The same is true for any other parameters that affect evaluation, e.g., the number of roll-outs per evaluation, the parameters of the environment, whether on-line or off-line evaluation is used, and the random seeds chosen.

**Collapsing across evaluation points.**   Some of the in-training reliability metrics (Dispersion across Runs, Risk across Runs, and Dispersion across Time) need to be evaluated at multiple evaluation

points along the training runs. If it is useful to obtain a small number of values to summarize each metric, we recommend dividing the training run into "time frames" (e.g. beginning, middle, and end), and collapsing across all evaluation points within each time frame.

**Normalization by performance.** Different algorithms can have vastly different ranges of performance even on the same task, and variability in performance tends to scale with actual performance. Thus, we normalize our metrics in post-processing by a measure of the range of performance for each algorithm. For "during training" reliability, we recommend normalizing by the median range of performance, which we define as the $p_{P_9 5} - p_{t=0}$, where $p_{P_9 5}$ is the 95th percentile and $p_{t=0}$ is the starting performance. For "after learning" reliability, the range of performance may not be available, in which case we use the median performance directly.

**Ranking the algorithms.** Because different environments have different ranges and distributions of reward, we must be careful when aggregating across environments or comparing between environments. Thus, if the analysis involves more than one environment, the per-environment median results for the algorithms are first converted to rankings, by ranking all algorithms within each task. To summarize the performance of a single algorithm across multiple tasks, we compute the mean ranking across tasks.

**Per-environment analysis.** The same algorithm can have different patterns of reliability for different environments. Therefore, we recommend inspecting reliability metrics on a per-environment basis, as well as aggregating across environments as described above.

## 4    CONFIDENCE INTERVALS AND STATISTICAL SIGNIFICANCE TESTS FOR COMPARISON

### 4.1    CONFIDENCE INTERVALS

We assume that the metric values have been converted to mean rankings, as explained in Section 3. To obtain confidence intervals on the mean rankings for each algorithm, we apply bootstrap sampling on the runs, by resampling runs with replacement (Efron & Tibshirani, 1986).

For metrics that are evaluated per-run (e.g. Dispersion across Time), we can resample the metric values directly, and then recompute the mean rankings on each resampling to obtain a distribution over the rankings; this allow us to compute confidence intervals. For metrics that are evaluated across-runs, we need to resample the runs themselves, then evaluate the metrics on each resampling, before recomputing the mean rankings to obtain a distribution on the mean rankings.

### 4.2    SIGNIFICANCE TESTS FOR COMPARING ALGORITHMS

Commonly, we would like to compare algorithms evaluated on a fixed set of environments. To determine whether any two algorithms have statistically significant differences in their metric rankings, we perform an exact permutation test on each pair of algorithms. Such tests allow us to compute a p-value for the null hypothesis (probability that the methods are in fact indistinguishable on the reliability metric).

We designed our permutation tests based on the null hypothesis that runs are exchangeable across the two algorithms being compared. In brief, let $A$ and $B$ be sets of performance measurements for algorithms $a$ and $b$. Let $Metric(X)$ be a reliability metric, e.g. the inter-quartile range across runs, computed on a set of measurements $X$. $MetricRanking(X)$ is the mean ranking across tasks on $X$, compared to the other algorithms being considered. We compute test statistic

$$s_{MetricRanking}(A, B) = MetricRanking(A) - MetricRanking(B).$$

Next we compute the distribution for $s_{MetricRanking}$ under the null hypothesis that the methods are equivalent, i.e. that performance measurements should have the same distribution for $a$ and $b$. We do this by computing random partitions $A', B'$ of $\{A \cup B\}$, and computing the test statistic $s_{MetricRanking}(A', B')$ on each partition. This yields a distribution for $s_{MetricRanking}$ (for sufficiently many samples), and the p-value can be computed from the percentile value of

$s_{MetricRanking}(A, B)$ in this distribution. As with the confidence intervals, a different procedure is required for per-run vs across-run metrics. Please see Appendix C for diagrams illustrating the permutation test procedures.

When performing pairwise comparisons between algorithms, it is critical to include corrections for multiple comparisons. This is because the probability of incorrect inferences increases with a greater number of simultaneous comparisons. We recommend using the Benjamini-Yekutieli method, which controls the false discovery rate (FDR), i.e., the proportion of rejected null hypotheses that are false.[4]

### 4.3 Reporting on statistical tests

It is important to report the details of any statistical tests performed, e.g. which test was used, the significance threshold, and the type of multiple-comparisons correction used.

## 5 Analysis of Reliability for Common Algorithms and Environments

In this section, we provide examples of applying the reliability metrics to a number of RL algorithms and environments, following the recommendations described above.

### 5.1 Continuous control algorithms on OpenAI Gym

We applied the reliability metrics to algorithms tested on seven continuous control environments from the Open-AI Gym (Greg Brockman et al., 2016) run on the MuJoCo physics simulator (Todorov et al., 2012). We tested REINFORCE (Sutton et al., 2000), DDPG (Lillicrap et al., 2015), PPO (Schulman et al., 2017), TD3 (Fujimoto et al., 2018), and SAC (Haarnoja et al., 2018) on the following Gym environments: Ant-v2, HalfCheetah-v2, Humanoid-v2, Reacher-v2, Swimmer-v2, and Walker2d-v2. We used the implementations of DDPG, TD3, and SAC from the TF-Agents library (Guadarrama et al., 2018). Each algorithm was run on each environment for 30 independent training runs.

We used a black-box optimizer (Golovin et al., 2017) to tune selected hyperparameters on a per-task basis, optimizing for final performance. The remaining hyperparameters were defined as stated in the corresponding original papers. See Appendix E for details of the hyperparameter search space and the final set of hyperparameters. During training, we evaluated the policies at a frequency of 1000 training steps. Each algorithm was run for a total of two million environment steps. For the "online" evaluations we used the generated training curves, averaging returns over recent training episodes collected using the exploration policy as it evolves. The raw training curves are shown in Appendix D. For evaluations after learning on a fixed policy, we took the last checkpoint from each training run as the fixed policy for evaluation. Each of these policies was then evaluated for 30 roll-outs, where each roll-out was defined as 1000 environment steps.

### 5.2 Discrete control: DQN variants on Atari

We also applied the reliability metrics to the RL algorithms and training data released as part of the Dopamine package (Castro et al., 2018). The data comprise the training runs of four RL algorithms, each applied to 60 Atari games. The RL algorithms are: DQN (Mnih et al., 2015), Implicit Quantile (IQN) (Dabney et al., 2018), C51 (Bellemare et al., 2017), and a variant of Rainbow implementing the three most important components (Hessel & Modayil, 2018). The algorithms were trained on each game for 5 training runs. Hyper-parameters follow the original papers, but were modified as necessary to follow Rainbow (Hessel & Modayil, 2018), to ensure apples-to-apples comparison. See Appendix E for the hyperparameters.

During training, the algorithms were evaluated in an "online" fashion every 1 million frames, averaging across the training episodes as recommended for evaluations on the ALE (Machado et al., 2018). Each training run consisted of approximately 200 million Atari frames (rounding to the nearest

---

[4]For situations in which a user wishes instead to control the family-wise error rate (FWER; the probability of incorrectly rejecting at least one true null hypothesis), we recommend using the Holm-Bonferroni method.

episode boundary every 1 million frames).[5] For evaluations after learning on a fixed policy ("after learning"), we took the last checkpoint from each training run as the fixed policies for evaluation. We then evaluated each of these policies for 125,000 environment steps.

## 5.3 PARAMETERS FOR RELIABILITY METRICS, CONFIDENCE INTERVALS, AND STATISTICAL TESTS

For the MuJoCo environments, we applied a sliding window of 100000 training steps for Dispersion across Time. For the Atari experiments, we used a sliding window size of 25 on top of the evaluations for the Dispersion across Time. For metrics with multiple evaluation points, we divided each training run into 3 time frames and averaged the metric rankings within each time frame. Because the results were extremely similar for all three time frames, we here report just for the final time frames.

Statistical tests for comparing algorithms were performed according to the recommendations in Section 4. We used pairwise permutation tests using 10,000 permutations per test, with a significance threshold of 0.05 and Benjamini-Yekutieli multiple-comparisons correction.

## 5.4 MEDIAN PERFORMANCE

The median performance of an algorithm is not a reliability metric, but it is interesting to see side-by-side with the reliability metrics. For analyzing median performance for the DQN variants, we used the normalization scheme of (Mnih et al., 2015), where an algorithm's performance is normalized against a lower baseline (e.g. the performance of a random policy) and an upper baseline (e.g. the performance of a human): $P_{\text{normalized}} = \frac{P - B_{\text{lower}}}{B_{\text{upper}} - B_{\text{lower}}}$. Median performance was not normalized for the continuous control algorithms.

## 5.5 RESULTS

The reliability metric rankings are shown in Fig. 1 for the MuJoCo results. We see that, according to Median Performance during training, SAC and TD3 have the best performance and perform similarly well, while REINFORCE performs the worst. However, SAC outperforms TD3 on all reliability metrics during training. Furthermore, both SAC and TD3 perform relatively poorly on all reliability metrics after learning, despite performing best on median performance.

The reliability metric rankings are shown in Fig. 2 for the Atari results. Here we see a similar result that, even though Rainbow performs significantly better than IQN in Median Performance, IQN performs numerically or significantly better than Rainbow on many of the reliability metrics.

The differing patterns in these metrics demonstrates that reliability is a separate dimension that needs to be inspected separately from mean or median performance – two algorithms may have similar median performance but may nonetheless significantly differ in reliability, as with SAC and TD3 above. Additionally, these results demonstrate that reliability along one axis does not necessarily correlate with reliability on other axes, demonstrating the value of evaluating these different dimensions so that algorithms can be compared and selected based on the requirements of the problem at hand.

To see metric results evaluated on a per-environment basis, please refer to Appendix F. Rank order of algorithms was often relatively consistent across the different environments evaluated. However, different environments did display different patterns across algorithms. For example, even though SAC showed the same or better Dispersion across Runs for most of the MuJoCo environments evaluated, it did show slightly worse Dispersion across Runs for the HalfCheetah environment (Fig 7a). This kind of result emphasizes the importance of inspecting reliability (and other performance metrics) on a per-environment basis, and also of evaluating reliability and performance on the environment of interest, if possible.

---

[5]The raw training curves can be viewed at https://google.github.io/dopamine/baselines/plots.html

## 6 CONCLUSION

We have presented a number of metrics, designed to measure different aspects of reliability of RL algorithms. We motivated the design goals and choices made in constructing these metrics, and also presented practical recommendations for the measurement of reliability for RL. Additionally, we presented examples of applying these metrics to common RL algorithms and environments, and showed that these metrics can reveal strengths and weaknesses of an algorithm that are obscured when we only inspect mean or median performance.

### ACKNOWLEDGMENTS

Many thanks to the following people for helpful discussions during the formulation of these metrics and the writing of the paper: Mohammad Ghavamzadeh, Yinlam Chow, Danijar Hafner, Rohan Anil, Archit Sharma, Vikas Sindhwani, Krzysztof Choromanski, Joelle Pineau, Hal Varian, Shyue-Ming Loh, and Tim Hesterberg. Thanks also to Toby Boyd for his assistance in the open-sourcing process, Oscar Ramirez for code reviews, and Pablo Castro for his help with running experiments using the Dopamine baselines data.

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

## A  ASSUMPTIONS AND DEFINITIONS

Reinforcement Learning algorithms vary widely in design, and our metrics are based on certain notions that should span the gamut of RL algorithms.

**Policy**  A policy $\pi_\Theta(a_i|s_i)$ is a distribution over actions $a_i$ given a current (input) state $s_i$. We assume policies are parameterized by a parameter $\Theta$.

**Agent**  An agent is defined as a distribution over policies (or equivalently a distribution over parameters $\Theta$). In many cases, an agent will be a single policy but for population-based RL methods, the agent is a discrete set of policies.

**Window**  A window is a collection of states over which the agent is assumed to have small variation. A window could be a sequence of consecutive time steps for a sequential RL algorithm, or a collection of states at the same training step of a distributed RL algorithm with a parameter server (all agents share $\Theta$).

**Performance**  The performance of an agent is the mean or median per-epoch reward from running that agent. If the agent is a single policy, then the performance $p(\pi_\Theta)$ is the mean or median per-epoch reward for that agent. If the agent is a distribution $D(\Theta)$ of policies, then the performance is the median of $p(\pi_\Theta)$ with $\Theta \sim D$.

**Training Run**  A training run is a sequence of updates to the agent $D(\Theta)$ from running a reinforcement learning algorithm. It leads to a trained agent $D_{final}(\Theta)$. Multiple training runs share no information with each other.

We cannot directly measure performance since it is a statistic across an infinite sample of evaluation runs of an agent. Instead we use windows to compute sample medians to approximate performance.

## B   DETRENDING BY DIFFERENCING

Typically, de-trending can be performed in two main ways (Nelson & Plosser, 1982; Hamilton, 1994). Differencing (i.e. $y_t\prime = y_t - y_{t-1}$) is more appropriate for difference-stationary (DS) processes (e.g. a random walk: $y_t = y_{t-1} + b + \epsilon_t$), where the shocks $\epsilon_t$ accumulate over time. For trend-stationary (TS) processes, which are characterized by stationary fluctuations around a deterministic trend, e.g. $y_t = a + b * t + \epsilon_t$, it is more appropriate to fit and subtract that trend.

We performed an analysis of real training runs and verified that the data are indeed approximately DS, and that differencing does indeed remove the majority of time-dependent structure. For this analysis we used the training runs on Atari as described in 5.2. Before differencing, the Augmented Dickey-Fuller test (ADF test, also known as a difference-stationarity test; Said E. Said & David A. Dickey (1984)) rejects the null hypothesis of a unit root on only 72% of the runs; after differencing, the ADF test rejects the null hypothesis on 92% of the runs (p-value threshold 0.05). For the ADF test, the rejection of a unit root (of the autoregressive lag polynomial) implies the alternate hypothesis, which is that the time series is trend-stationary.

Therefore, our training curves are better characterized as an accumulation of shocks, i.e. as DS processes, rather than as mean-reverting TS processes. They are not actually purely DS because the shocks $\epsilon_t$ are not stationary over time, but because we compute standard deviation within sliding windows, we can capture the non-stationarity and change in variability over time. Thus, we chose to detrend using differencing.

As a further note in favor of detrending by differencing, it is useful to observe that many measures of variability are defined relative to the central tendency of the data, e.g. the median absolute deviation $\text{MAD} = \text{median}(|X_i - \widetilde{X}|)$ where $\widetilde{X}$ is the median of $X$. On the raw data (without differencing), the MAD would be defined relative to $\widetilde{X}$ as median performance, so that any improvements in performance are included in that computation of variability. On the other hand, if we compute MAD on the 1st-order differences, we are using a $\widetilde{X}$ that represents the median *change* in performance, which is a more reasonable baseline to compute variability against, when we are in fact concerned with the variability of those changes.

A final benefit of differencing is that it is parameter-free.

## C   ILLUSTRATIONS OF PERMUTATION TEST PROCEDURES

We illustrate the procedure for computing permutation tests to compare pairs of algorithms on a specified metric, in Figs. 3 (for per-run metrics) and 4 (for across-run metrics).

## D   RAW TRAINING CURVES FOR OPENAI MUJOCO TASKS

In Figure 5, we show the raw training curves for the TF-Agents implementations of continuous-control algorithms, applied to the OpenAI MuJoCo tasks. These are compared against baselines from the literature, where available (DDPG and TD3: Fujimoto et al. (2018), PPO: Schulman et al. (2017), SAC: Haarnoja et al. (2018))

## E   HYPERPARAMETER SETTINGS

For the continuous control experiments, hyperparameters were chosen on a per-environment basis according to the black-box optimization algorithm described in Golovin et al. (2017). The hyperparameter search space is shown in Table 2.

For the discrete control experiments, hyperparameter selection is described in (Castro et al., 2018). Hyperparameters are shown in Table 8, duplicated for reference from https://github.com/google/dopamine/tree/master/baselines.

**Comparing algorithms on per-run metrics**

**Raw values**  *e.g. 3 runs per (task, algo)*

| | task1 | task2 | task3 |
|---|---|---|---|
| algoA | -1, -7, 3 | 2.5, 7, 3 | 77, 90, 4 |
| algoB | -4, 2, 0 | 1.9, 0.3, 4 | 5, 32, 15 |
| algoC | 3, 2, 4 | 6, 10, 5 | 52, 64, 3 |

*Evaluate per-run metrics for each run*

**Metric values**

| | | |
|---|---|---|
| 0.1, 0.7, 3 | 2.3, 4.1, 3 | 3, 6, 9 |
| -.4, .9, 1 | 2, 2.5, 1.2 | 0.6, 1, 4 |
| 0, -0.2, 4 | 0.9, 0, 1.6 | 8, -2, 0.3 |

algorithms / tasks

*Rank within each task*

**Metric rankings**

| | | |
|---|---|---|
| 4, 5, 8 | 6, 9, 8 | 5, 7, 9 |
| 1, 6, 7 | 5, 7, 3 | 3, 4, 6 |
| 3, 2, 9 | 2, 1, 4 | 8, 1, 2 |

algorithms / tasks

*Permute rankings across algoA and algoB, within task*

**Permuted rankings**

| | | |
|---|---|---|
| 1, 6, 7 | 7, 5, 3 | 3, 7, 9 |
| 5, 8, 4 | 6, 9, 8 | 5, 6, 4 |
| 3, 2, 9 | 2, 1, 4 | 8, 1, 2 |

permuted

algorithms / tasks

*Get difference in mean ranking: algoA - algoB = 6.8 - 4.7*

*Get difference in mean ranking: algoA - algoB = 5.3 - 6.0*

*Repeat n_permutations times*

null distribution

actual difference

Figure 3: Diagram illustrating the computation of the permutation tests for **per-run metrics** (Dispersion across Time, Short-term Risk across Time, Long-term Risk across Time). In this example, we are comparing Algorithm A and Algorithm B, and there are only 3 algorithms, 3 tasks, and 3 runs per (task, algo) pair. To compute the difference in average rankings for two algorithms, follow the gray arrows. To compute a null distribution of difference in average rankings (by permuting the runs), follow the blue arrows a number of times (e.g. 1,000 times). Once the null distribution has been computed, the actual value of the difference can be compared with the null distribution to obtain a p-value.

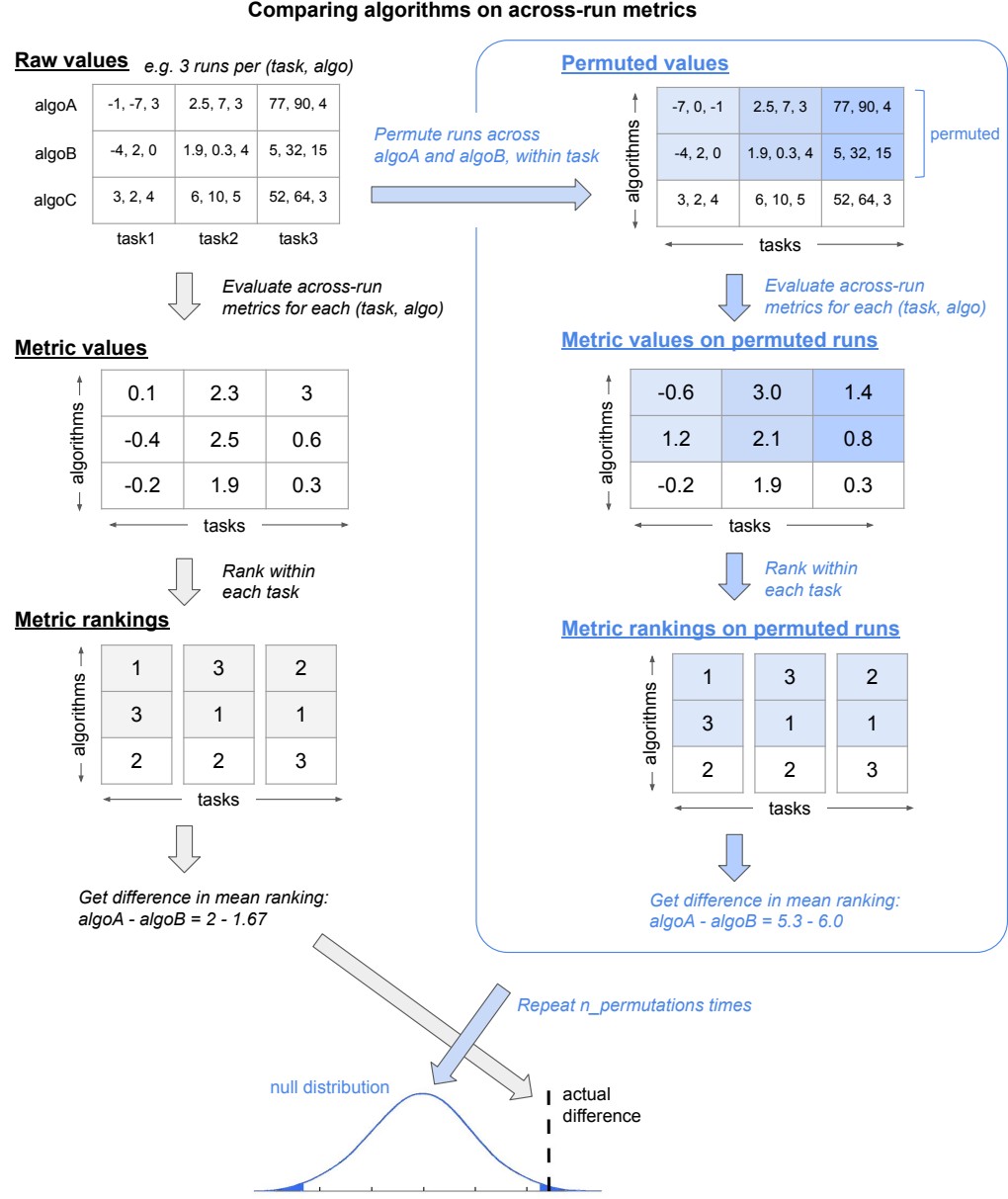

Figure 4: Diagram illustrating the computation of the permutation tests for **across-run or across-rollout metrics** (Dispersion across Runs, Risk Across Runs, Dispersion across Fixed-policy rollouts, Risk across Fixed-Policy rollouts). In this example, we are comparing Algorithm A and Algorithm B, and there are only 3 algorithms, 3 tasks, and 3 runs per (task, algo) pair. To compute the difference in average rankings for two algorithms, follow the gray arrows. To compute a null distribution of difference in average rankings (by permuting the runs), follow the blue arrows a number of times (e.g. 1,000 times). Once the null distribution has been computed, the actual value of the difference can be compared with the null distribution to obtain a p-value.

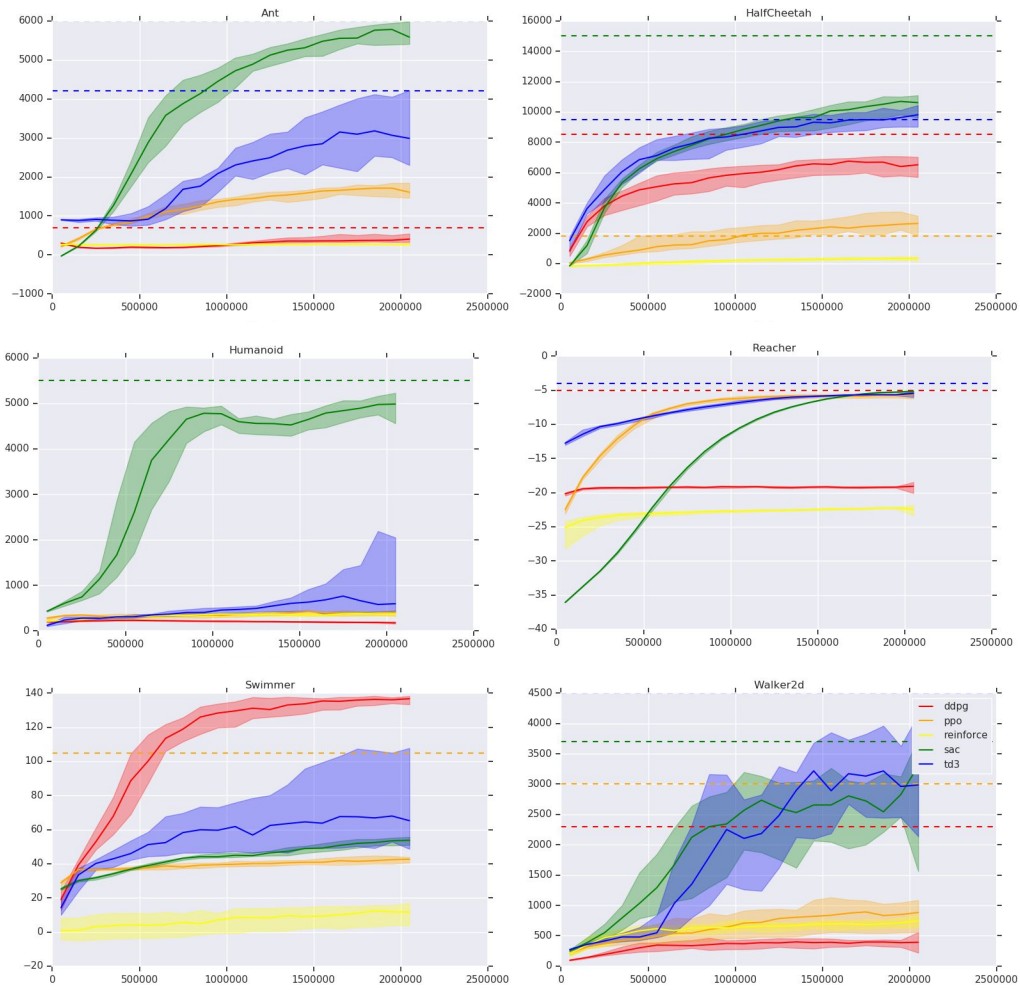

Figure 5: Raw training curves for OpenAI MuJoCo tasks. The x-axes indicate environment steps, and the y-axes indicate average per-episode return. Dotted lines indicate baseline performance from the literature, where available.

Table 2: Hyperparameter search space for continuous control algorithms.

| Algorithm | Hyperparameter | Search min | Search max |
|---|---|---|---|
| SAC | actor learning rate | 0.000001 | 0.001 |
| | $\alpha$ learning rate | 0.000001 | 0.001 |
| | critic learning rate | 0.000001 | 0.001 |
| | target update $\tau$ | 0.00001 | 1.0 |
| TD3 | actor learning rate | 0.000001 | 0.001 |
| | critic learning rate | 0.000001 | 0.001 |
| | target update $\tau$ | 0.00001 | 1.0 |
| PPO | learning rate | 0.000001 | 0.001 |
| DDPG | actor learning rate | 0.000001 | 0.001 |
| | critic learning rate | 0.000001 | 0.001 |
| | target update $\tau$ | 0.00001 | 1.0 |
| REINFORCE | learning rate | 0.000001 | 0.001 |
| | # episodes before each train step | 1.0 | 10 |

Table 3: Final hyperparameters for SAC.

| | actor learning rate | $\alpha$ learning rate | critic learning rate | target update $\tau$ |
|---|---|---|---|---|
| Ant-v2 | 0.000006 | 0.000009 | 0.0009 | 0.0002 |
| HalfCheetah-v2 | 0.0001 | 0.000005 | 0.0004 | 0.02 |
| Humanoid-v2 | 0.0003 | 0.0008 | 0.0006 | 0.8 |
| Reacher-v2 | 0.00001 | 0.000002 | 0.0005 | 0.00002 |
| Swimmer-v2 | 0.000004 | 0.000009 | 0.0002 | 0.009 |
| Walker2d-v2 | 0.0002 | 0.0009 | 0.0008 | 0.01 |

Table 4: Final hyperparameters for TD3.

| | actor learning rate | critic learning rate | target update $\tau$ |
|---|---|---|---|
| Ant-v2 | 0.000001 | 0.0002 | 0.0003 |
| HalfCheetah-v2 | 0.0003 | 0.0005 | 0.02 |
| Humanoid-v2 | 0.0001 | 0.0001 | 0.0002 |
| Reacher-v2 | 0.000001 | 0.00003 | 0.00003 |
| Swimmer-v2 | 0.0004 | 0.0002 | 0.01 |
| Walker2d-v2 | 0.00006 | 0.00009 | 0.001 |

Table 5: Final hyperparameters for PPO.

| | learning rate |
|---|---|
| Ant-v2 | 0.0008 |
| HalfCheetah-v2 | 0.0008 |
| Humanoid-v2 | 0.0008 |
| Reacher-v2 | 0.00002 |
| Swimmer-v2 | 0.0004 |
| Walker2d-v2 | 0.0002 |

Table 6: Final hyperparameters for DDPG.

|  | actor learning rate | critic learning rate | target update $\tau$ |
|---|---|---|---|
| Ant-v2 | 0.00003 | 0.0004 | 0.0002 |
| HalfCheetah-v2 | 0.00006 | 0.0005 | 0.02 |
| Humanoid-v2 | 0.00006 | 0.00009 | 0.01 |
| Reacher-v2 | 0.00005 | 0.0005 | 0.005 |
| Swimmer-v2 | 0.0005 | 0.0003 | 0.004 |
| Walker2d-v2 | 0.0003 | 0.0004 | 0.03 |

Table 7: Final hyperparameters for REINFORCE.

|  | learning rate | # episodes before each train step |
|---|---|---|
| Ant-v2 | 0.00002 | 9 |
| HalfCheetah-v2 | 0.0004 | 7 |
| Humanoid-v2 | 0.0005 | 2 |
| Reacher-v2 | 0.000004 | 6 |
| Swimmer-v2 | 0.000005 | 3 |
| Walker2d-v2 | 0.0001 | 6 |

Table 8: Hyperparameters for discrete control algorithms.

| Training $\epsilon$ | Evaluation $\epsilon$ | $\epsilon$ decay schedule | Min. history to start learning | Target network update frequency |
|---|---|---|---|---|
| 0.01 | 0.001 | 1,000,000 frames | 80,000 frames | 32,000 frames |

## F PER-TASK METRIC RESULTS

Metric results are shown on a per-task basis in Figs. 6 to 8 for the OpenAI Gym MuJoCo tasks, and Figs. 9 to 23 for the Atari environments. Note that because we are no longer aggregating across tasks in this analysis, we do not need to convert the metric values to rankings.

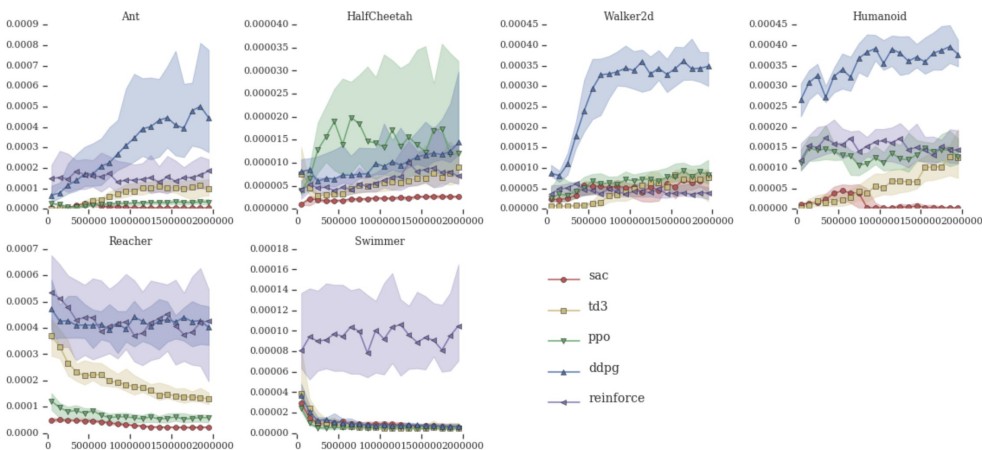

(a) Dispersion across Time. Better reliability is indicated by less positive values. The x-axes indicate the number of environment steps.

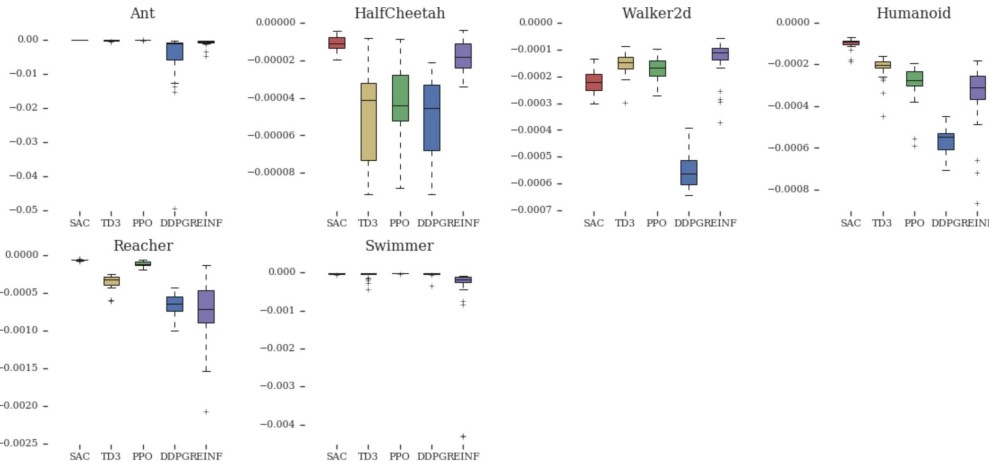

(b) Short-term Risk across Time. Better reliability is indicated by more positive values.

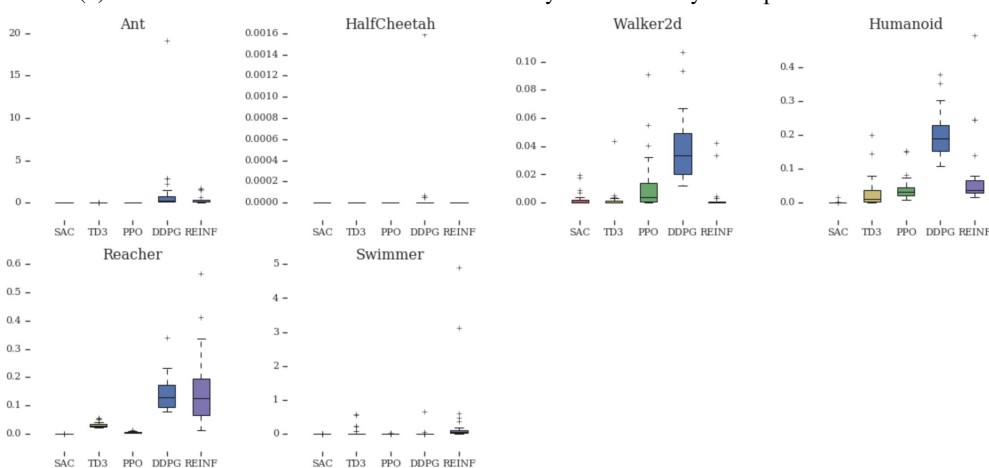

(c) Long-term Risk across Time. Better reliability is indicated by less positive values.

Figure 6: Across-time reliability metrics for continuous control RL algorithms tested on OpenAI Gym environments, evaluated on a per-environment basis.

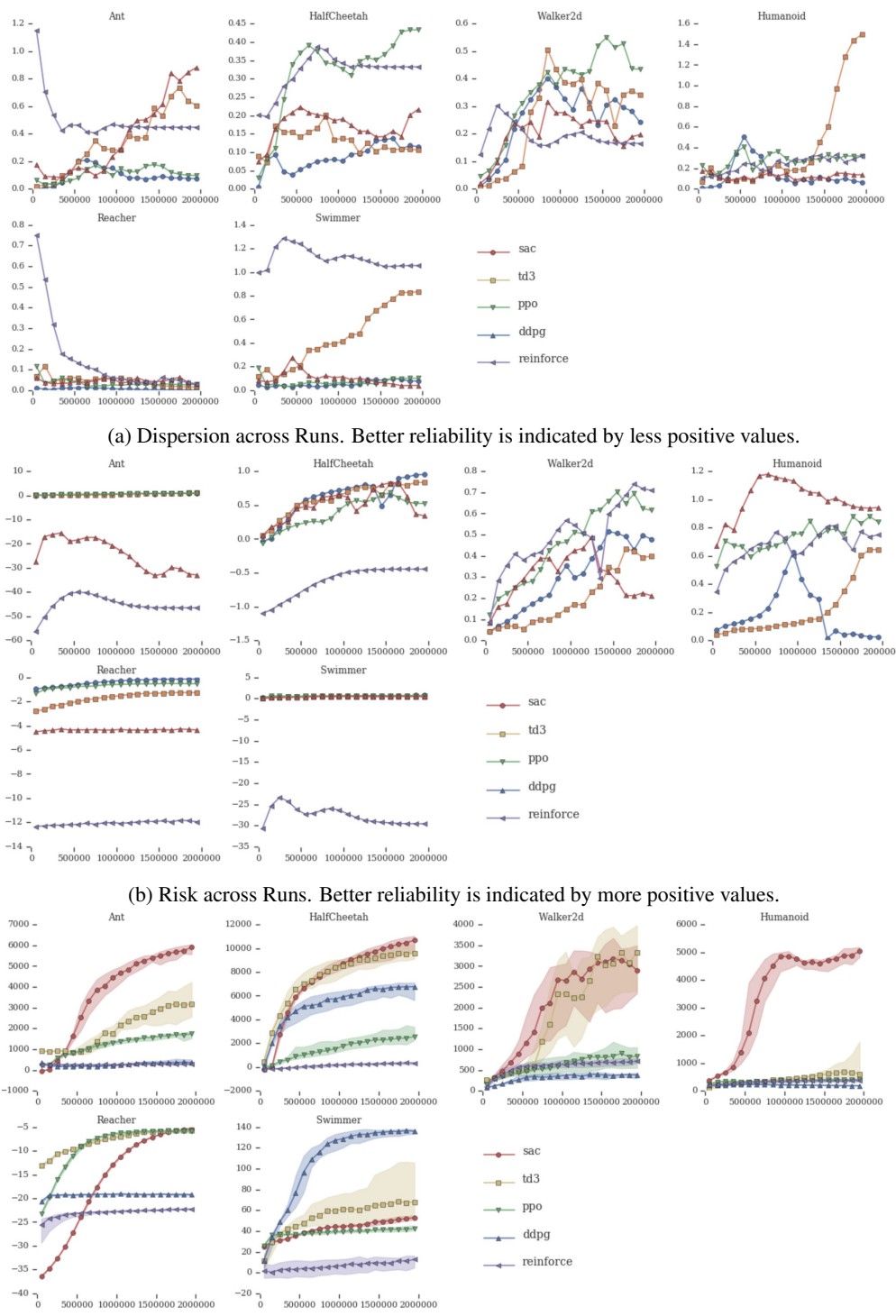

(a) Dispersion across Runs. Better reliability is indicated by less positive values.

(b) Risk across Runs. Better reliability is indicated by more positive values.

(c) Median performance during training. Better performance is indicated by more positive values.

Figure 7: Across-run reliability metrics and median performance for continuous control RL algorithms tested on OpenAI Gym environments, evaluated on a per-environment basis. The x-axes indicate the number of environment steps.

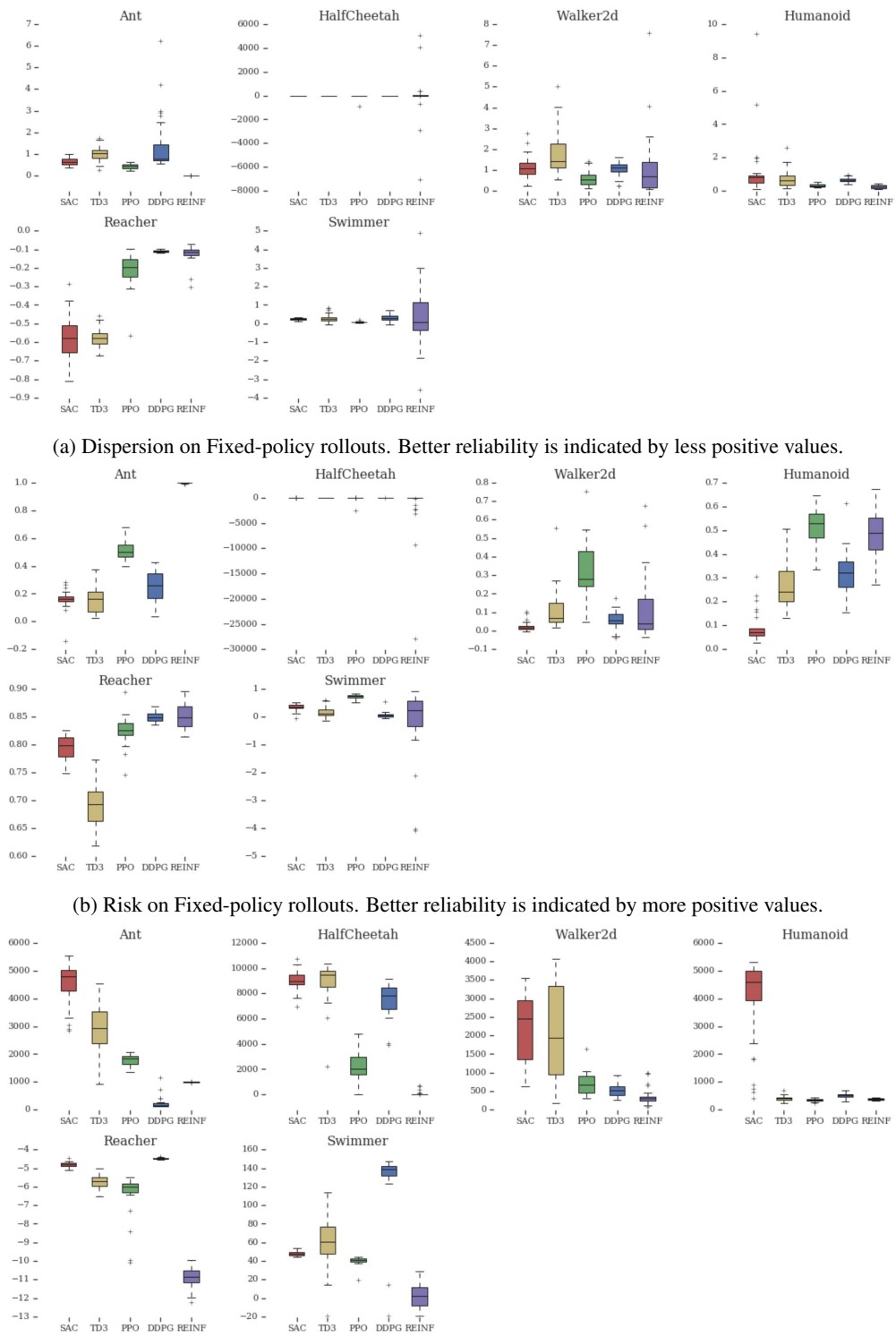

(a) Dispersion on Fixed-policy rollouts. Better reliability is indicated by less positive values.

(b) Risk on Fixed-policy rollouts. Better reliability is indicated by more positive values.

(c) Median performance on Fixed-policy rollouts. Better performance is indicated by more positive values.

Figure 8: Reliability metrics and median performance on fixed-policy rollouts for continuous control RL algorithms tested on OpenAI Gym environments, evaluated on a per-environment basis.

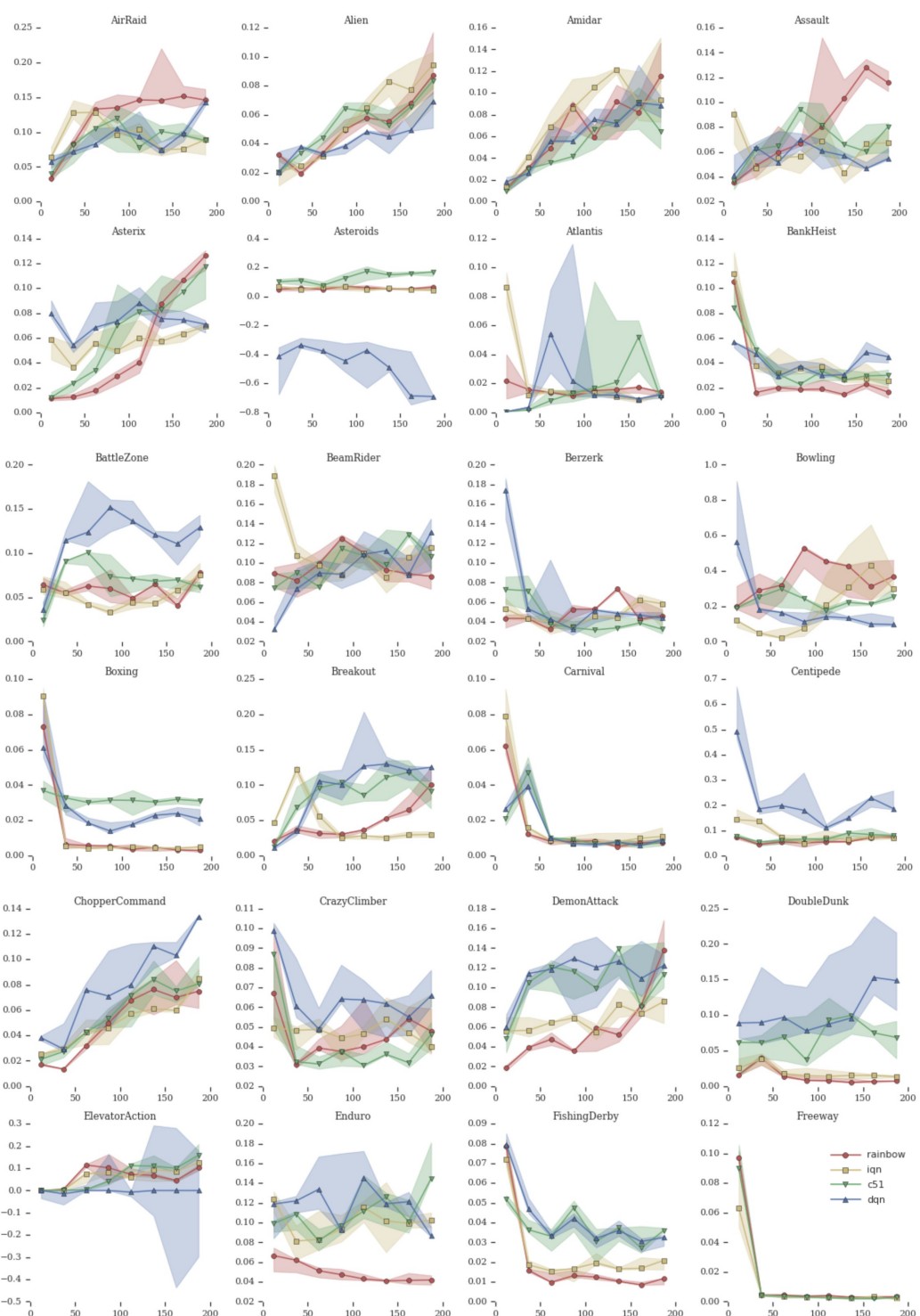

Figure 9: Dispersion across Time for DQN-variants tested on 60 Atari games, evaluated on a per-environment basis (page 1). Better reliability is indicated by less positive values. The x-axes indicate millions of Atari frames.

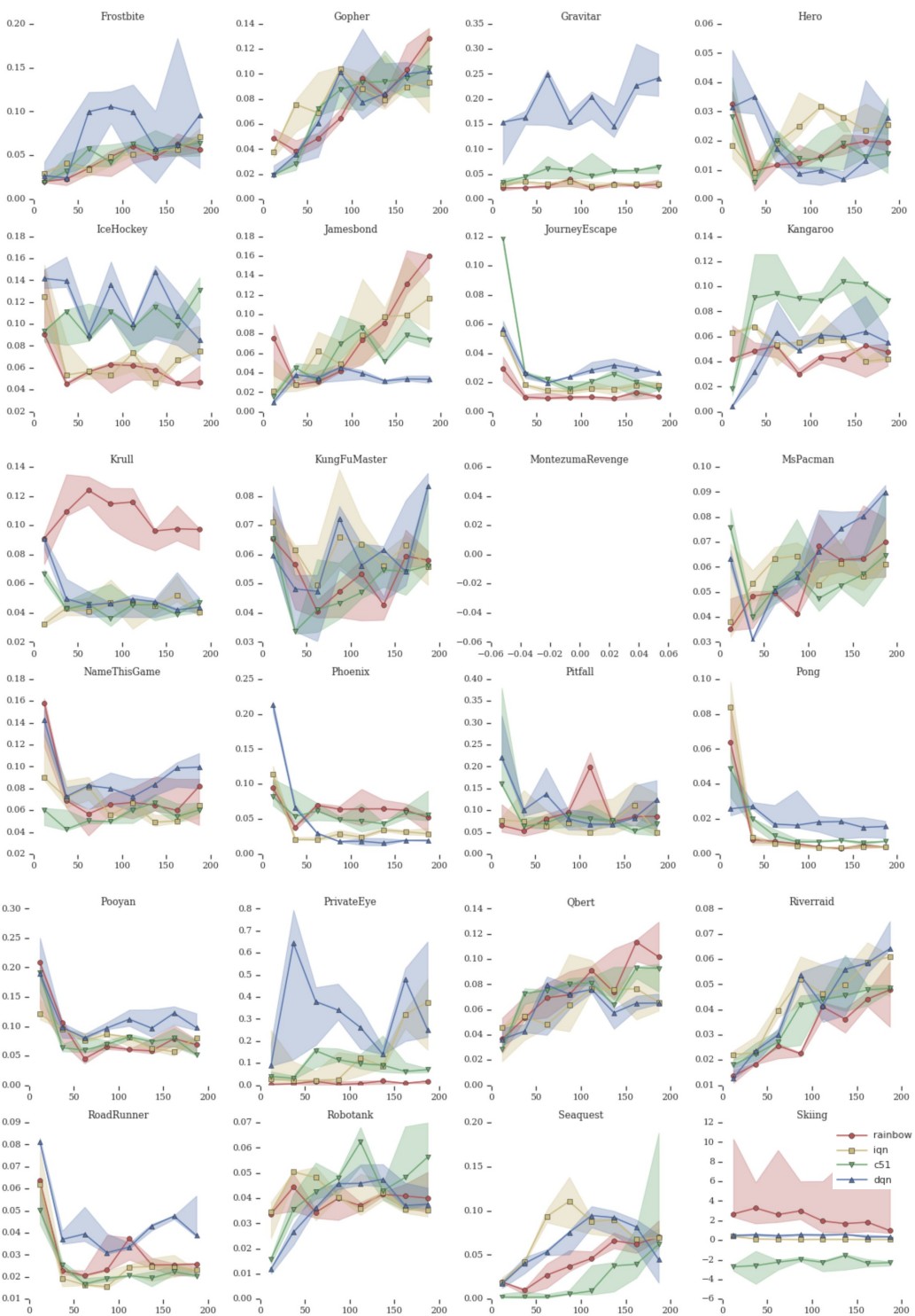

Figure 10: Dispersion across Time for DQN-variants tested on 60 Atari games, evaluated on a per-environment basis (page 2). Better reliability is indicated by less positive values. The x-axes indicate millions of Atari frames.

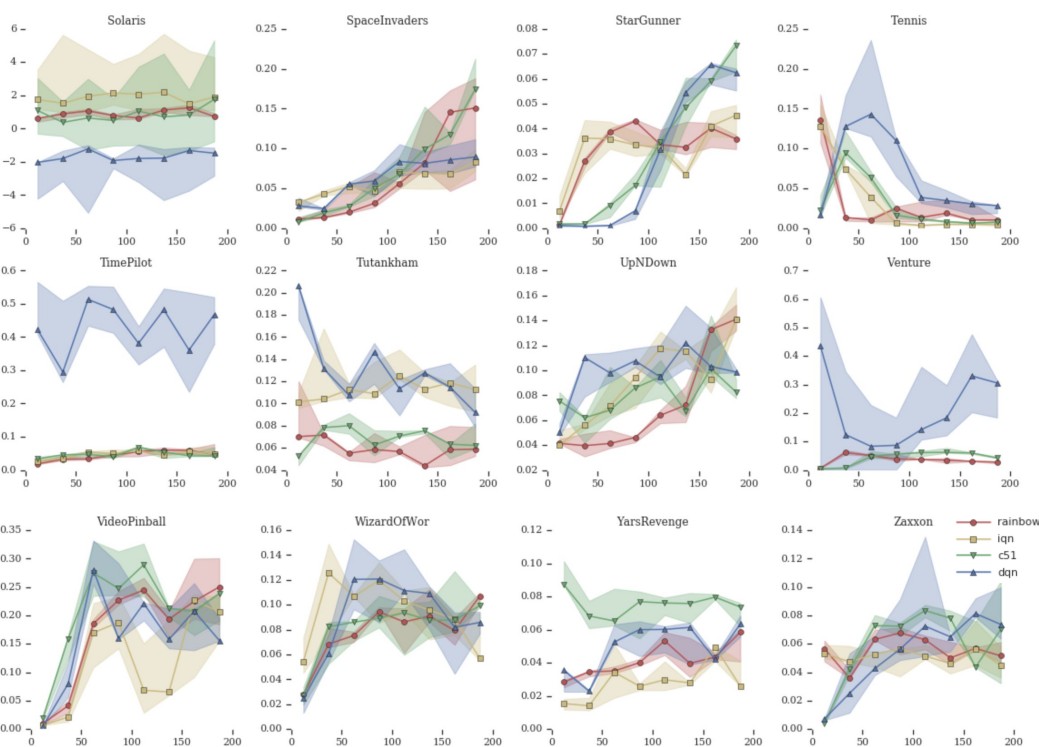

Figure 11: Dispersion across Time for DQN-variants tested on 60 Atari games, evaluated on a per-environment basis (page 3). Better reliability is indicated by less positive values. The x-axes indicate millions of Atari frames.

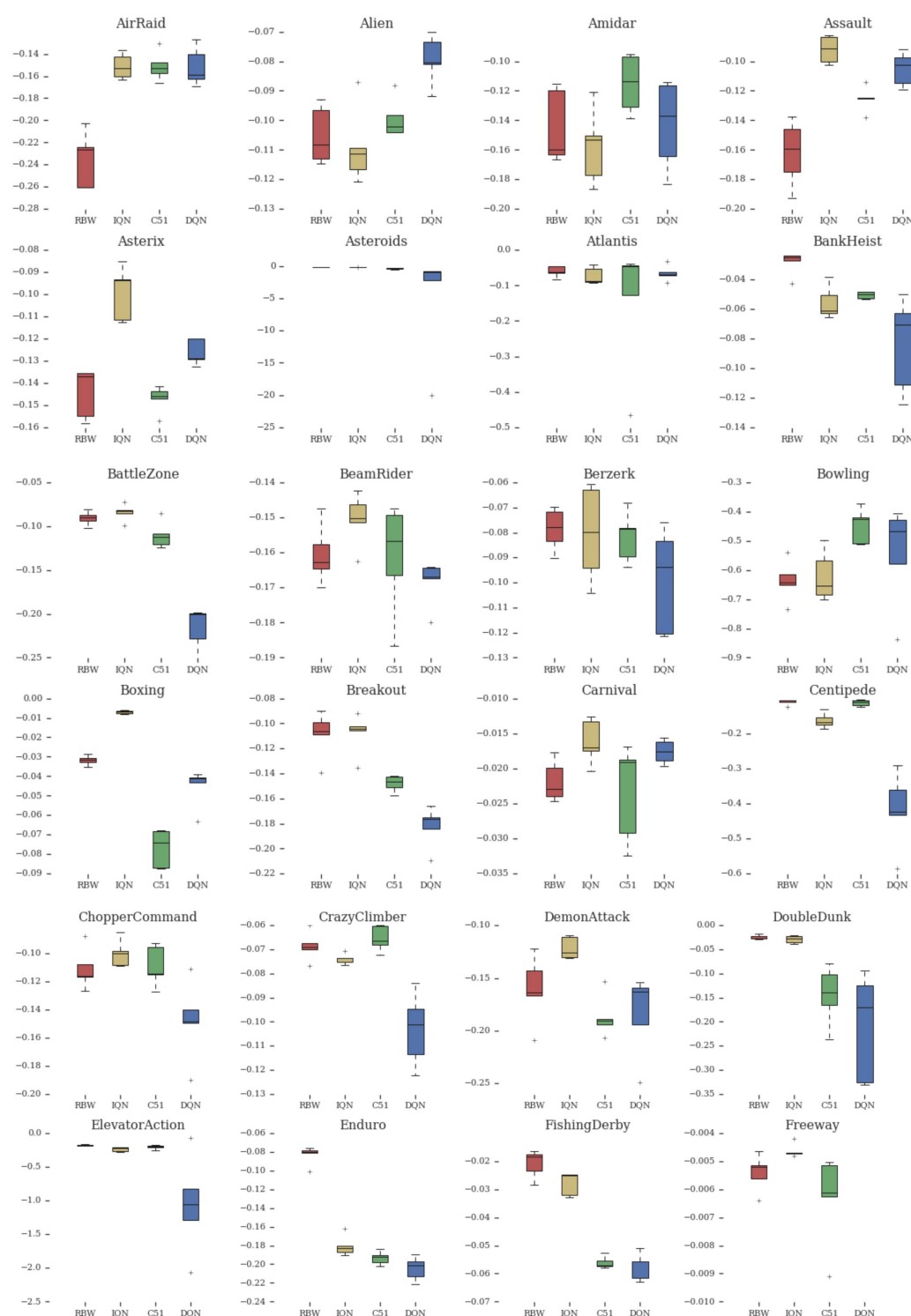

Figure 12: Short-term Risk across Time for DQN-variants tested on 60 Atari games, evaluated on a per-environment basis (page 1). Better reliability is indicated by less positive values. The x-axes indicate millions of Atari frames.

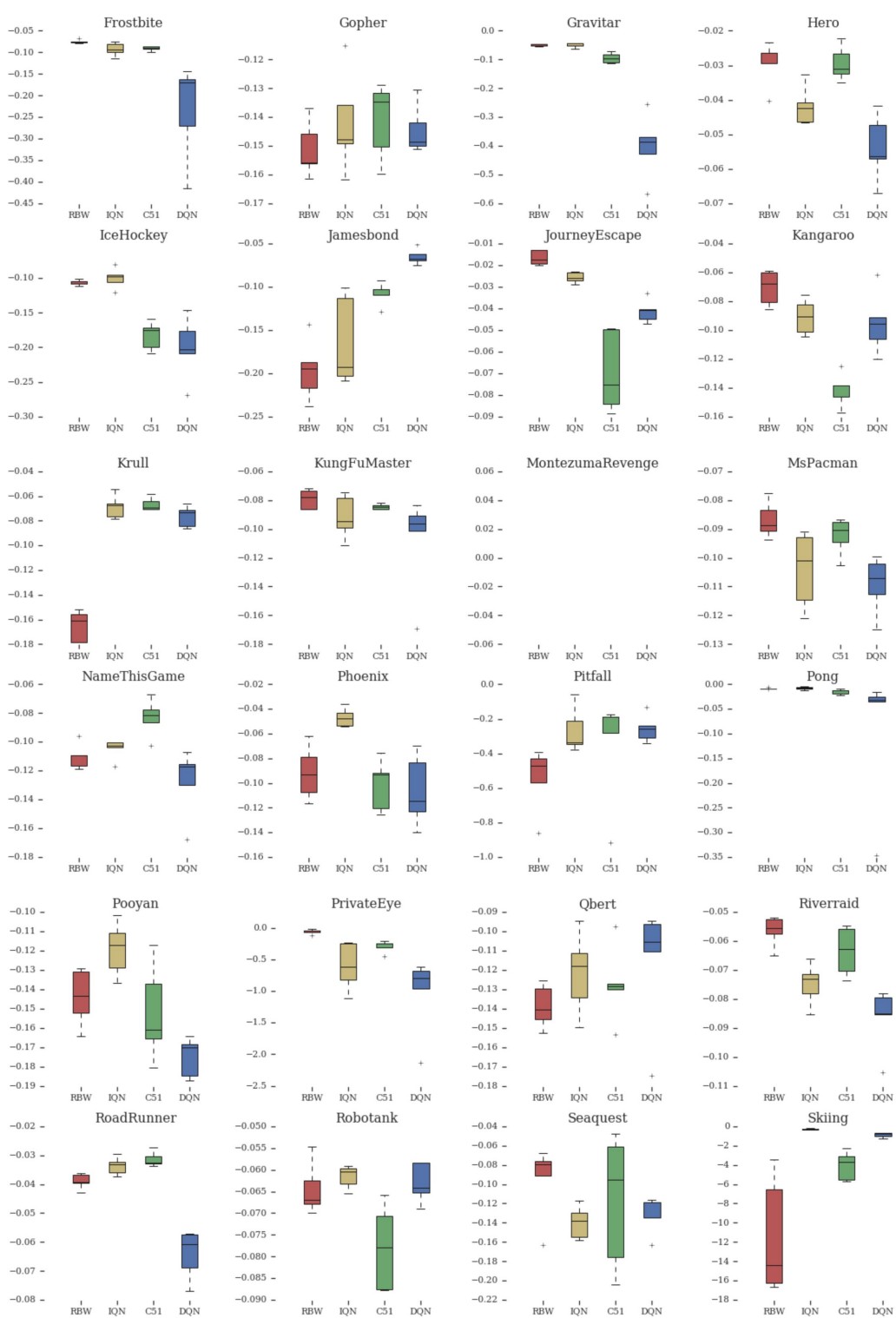

Figure 13: Short-term Risk across Time for DQN-variants tested on 60 Atari games, evaluated on a per-environment basis (page 2). Better reliability is indicated by less positive values. The x-axes indicate millions of Atari frames.

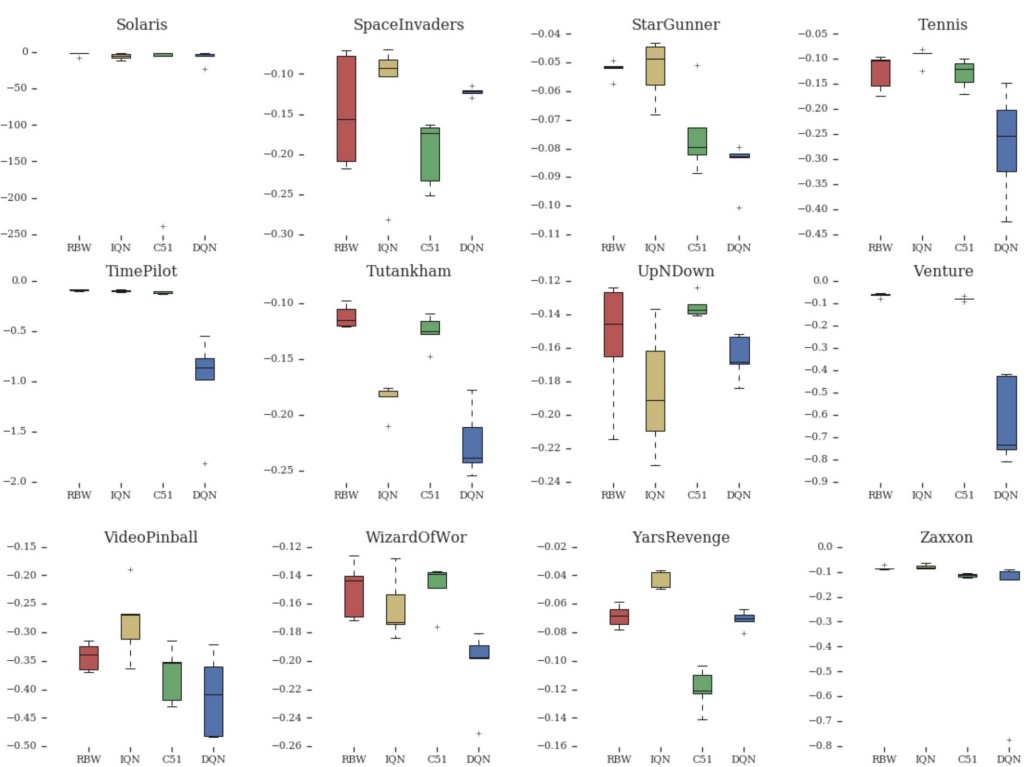

Figure 14: Short-term Risk across Time for DQN-variants tested on 60 Atari games, evaluated on a per-environment basis (page 3). Better reliability is indicated by less positive values. The x-axes indicate millions of Atari frames.

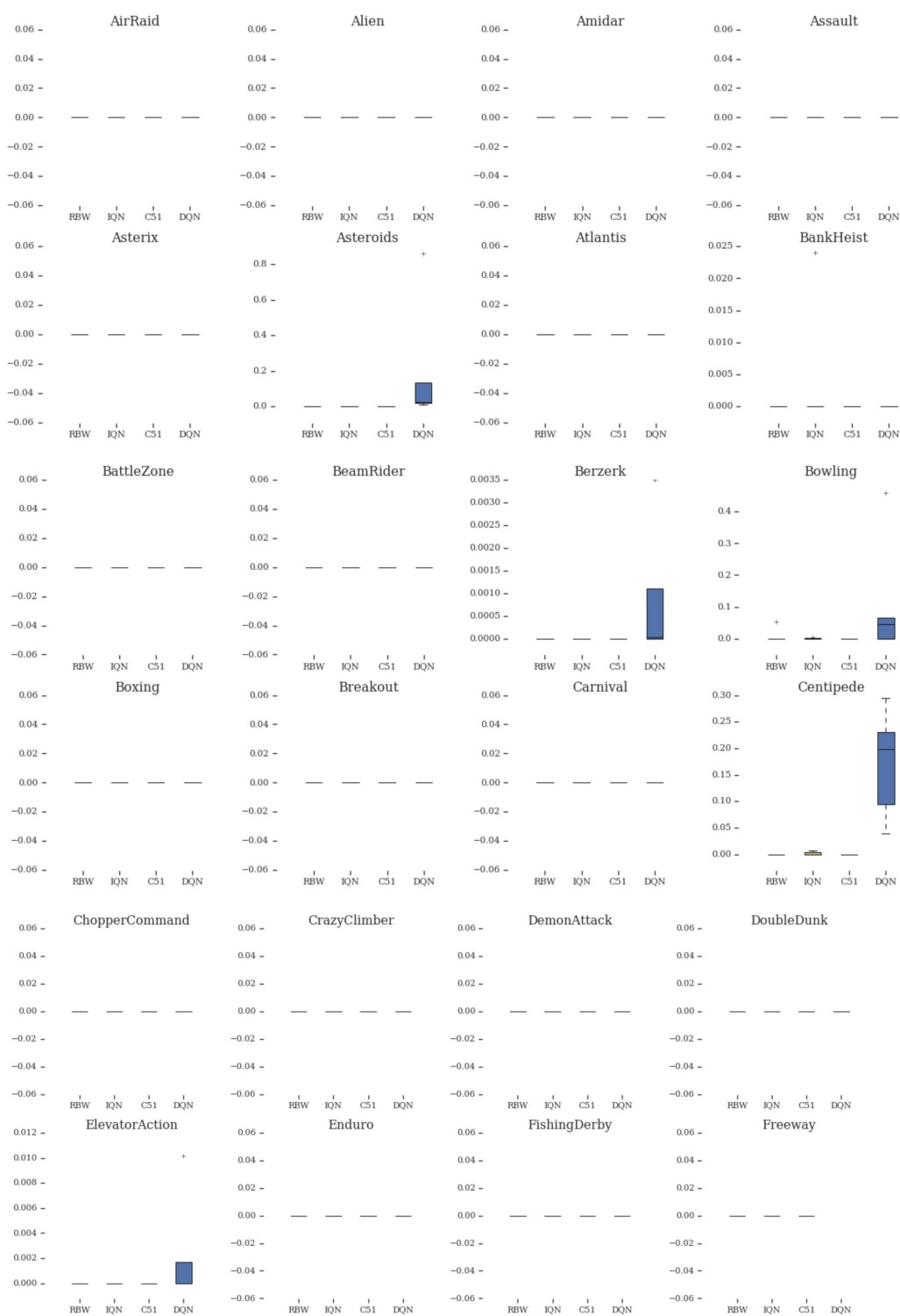

Figure 15: Long-term Risk across Time for DQN-variants tested on 60 Atari games, evaluated on a per-environment basis (page 1). Better reliability is indicated by less positive values. The x-axes indicate millions of Atari frames.

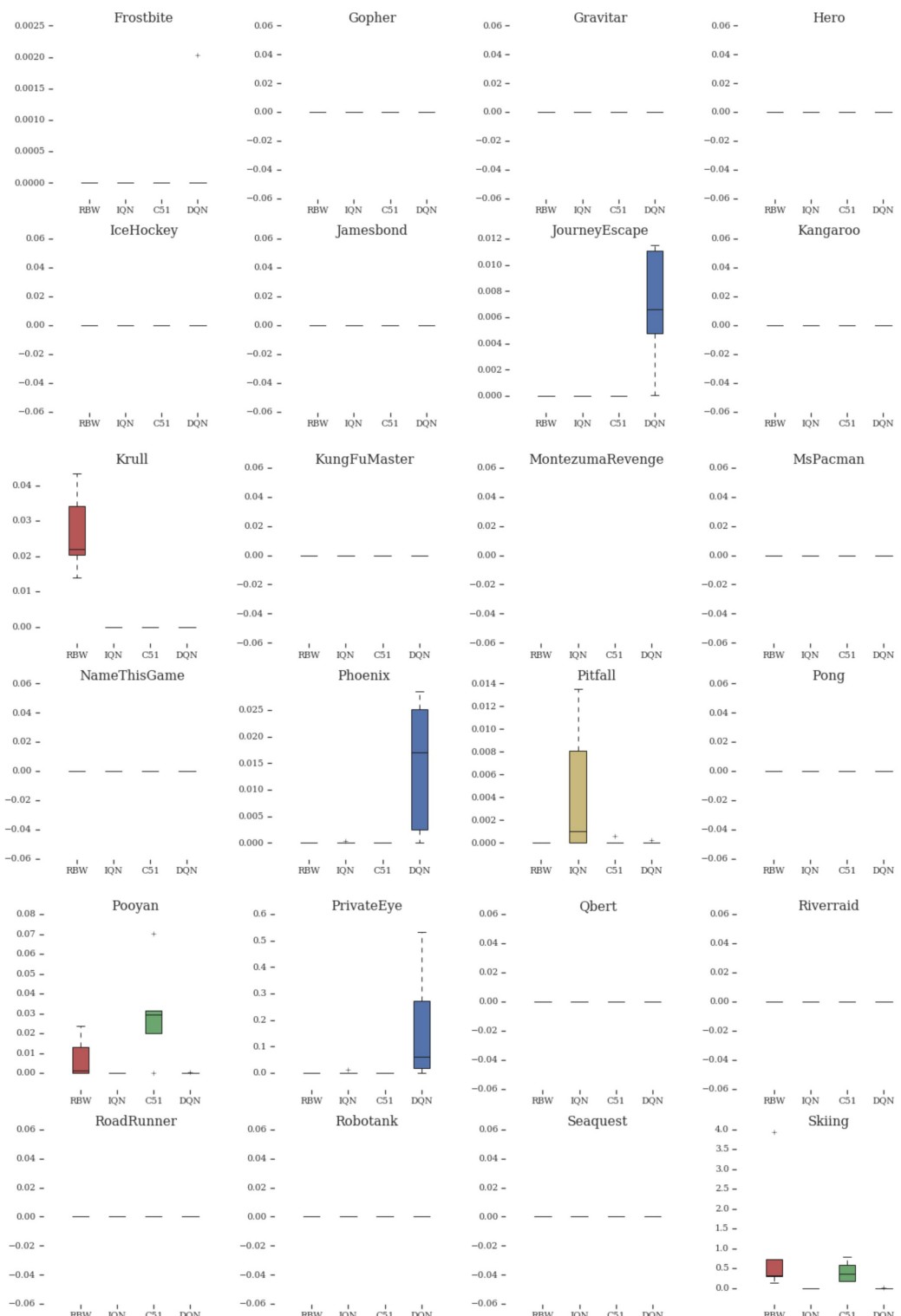

Figure 16: Long-term Risk across Time for DQN-variants tested on 60 Atari games, evaluated on a per-environment basis (page 2). Better reliability is indicated by less positive values. The x-axes indicate millions of Atari frames.

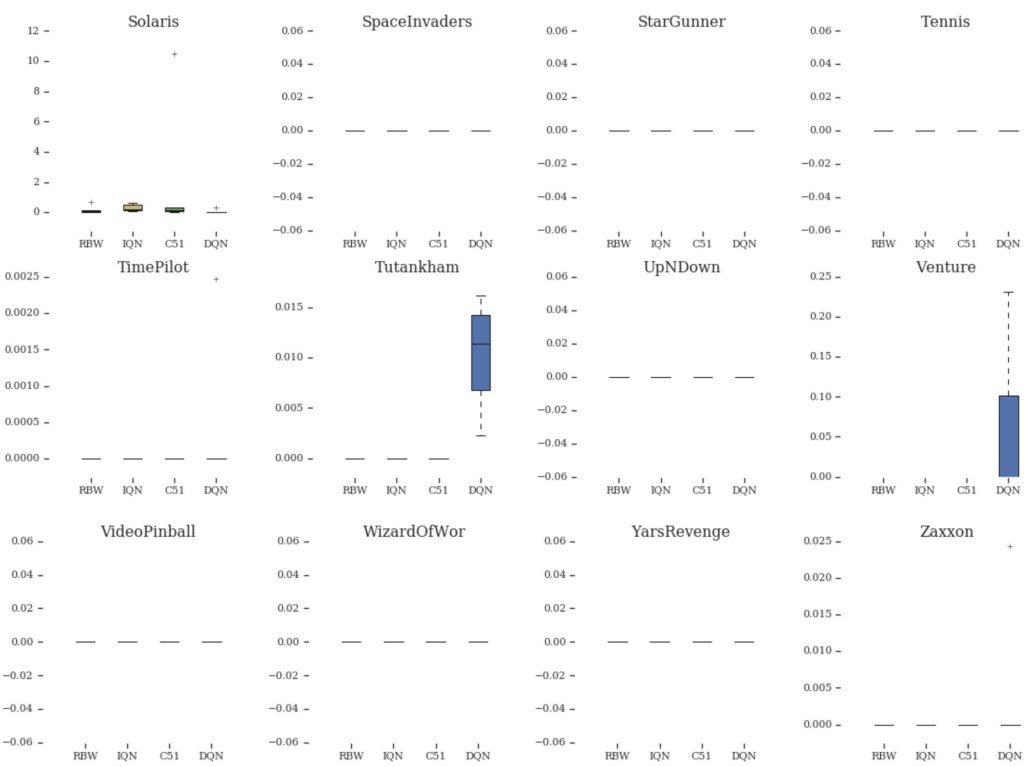

Figure 17: Long-term Risk across Time for DQN-variants tested on 60 Atari games, evaluated on a per-environment basis (page 3). Better reliability is indicated by less positive values. The x-axes indicate millions of Atari frames.

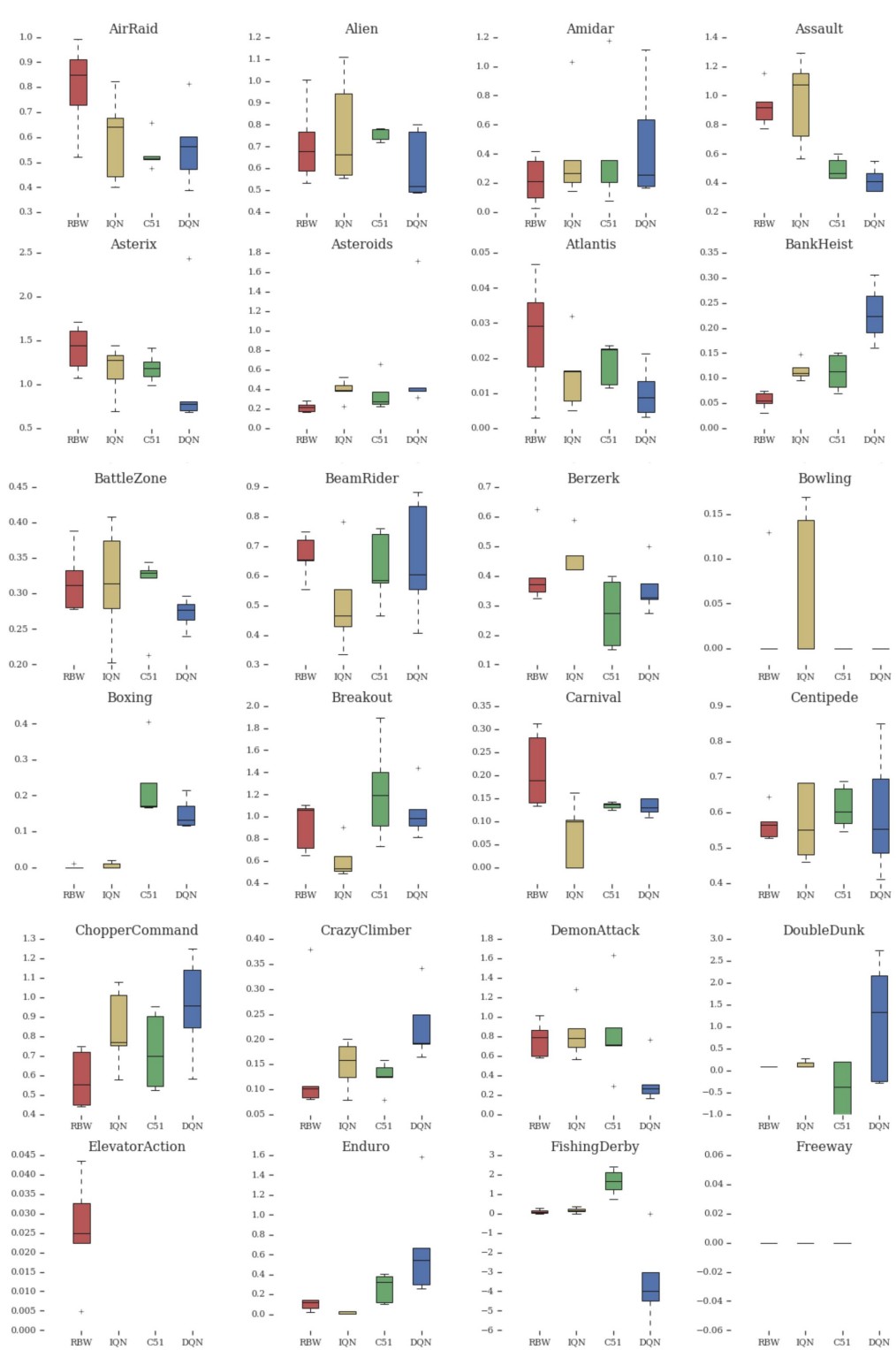

Figure 18: Dispersion across Fixed-policy Rollouts for DQN-variants tested on 60 Atari games, evaluated on a per-environment basis (page 1). Better reliability is indicated by less positive values. The x-axes indicate millions of Atari frames.

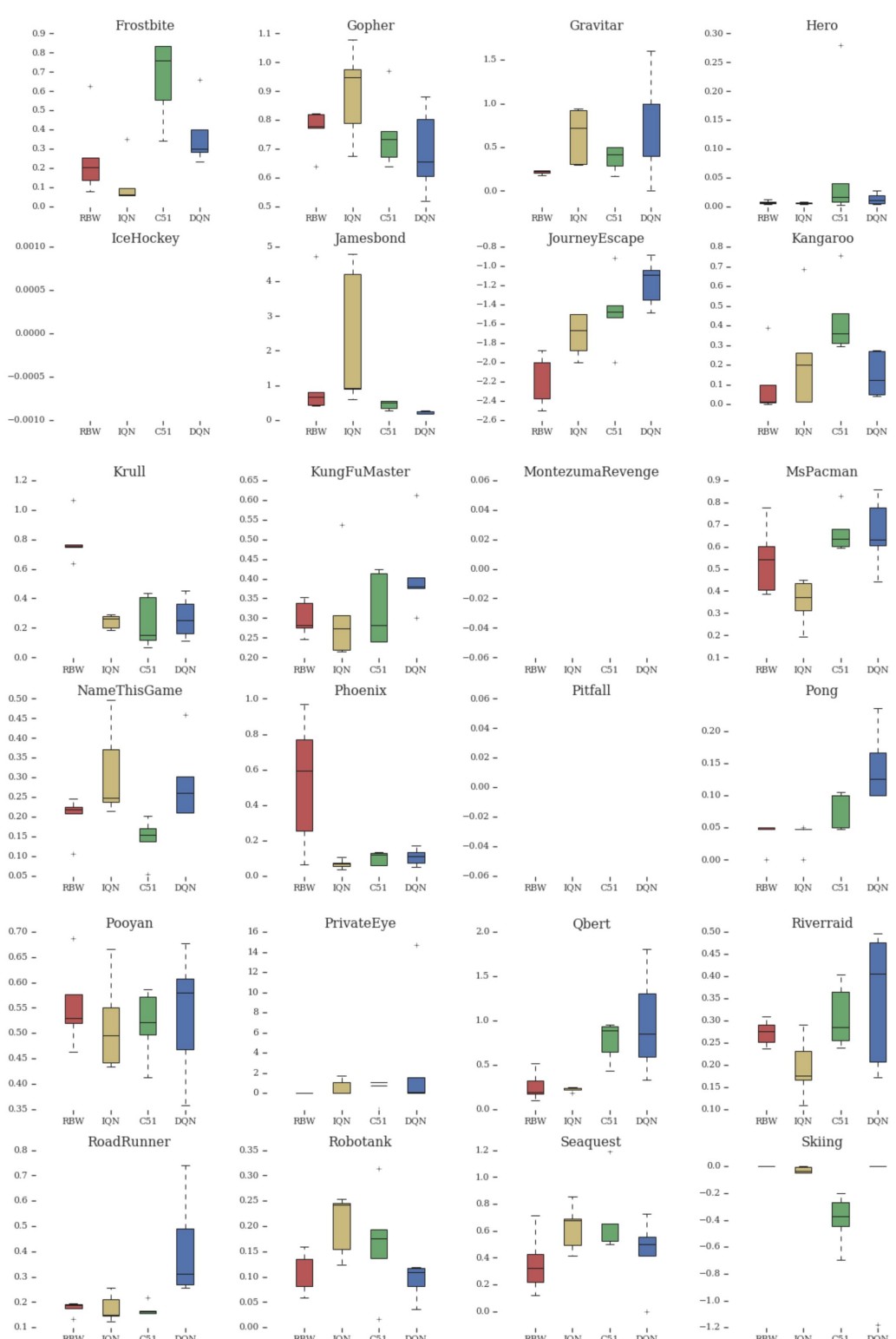

Figure 19: Dispersion across Fixed-policy Rollouts for DQN-variants tested on 60 Atari games, evaluated on a per-environment basis (page 2). Better reliability is indicated by less positive values. The x-axes indicate millions of Atari frames.

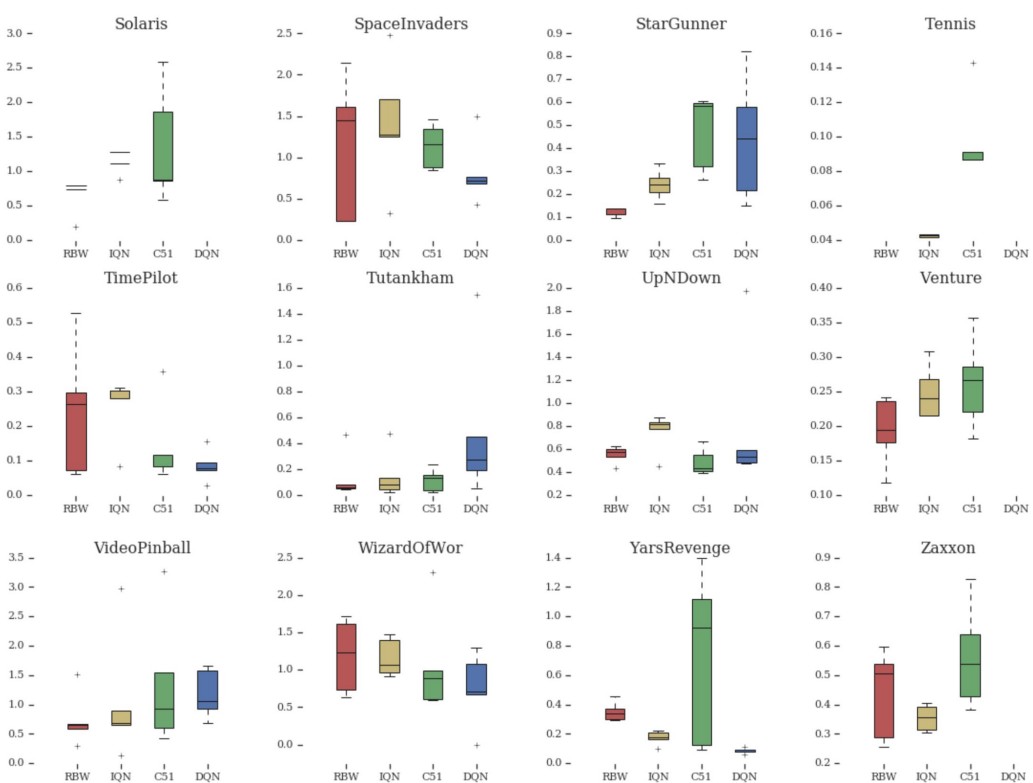

Figure 20: Dispersion across Fixed-policy Rollouts for DQN-variants tested on 60 Atari games, evaluated on a per-environment basis (page 3). Better reliability is indicated by less positive values. The x-axes indicate millions of Atari frames.

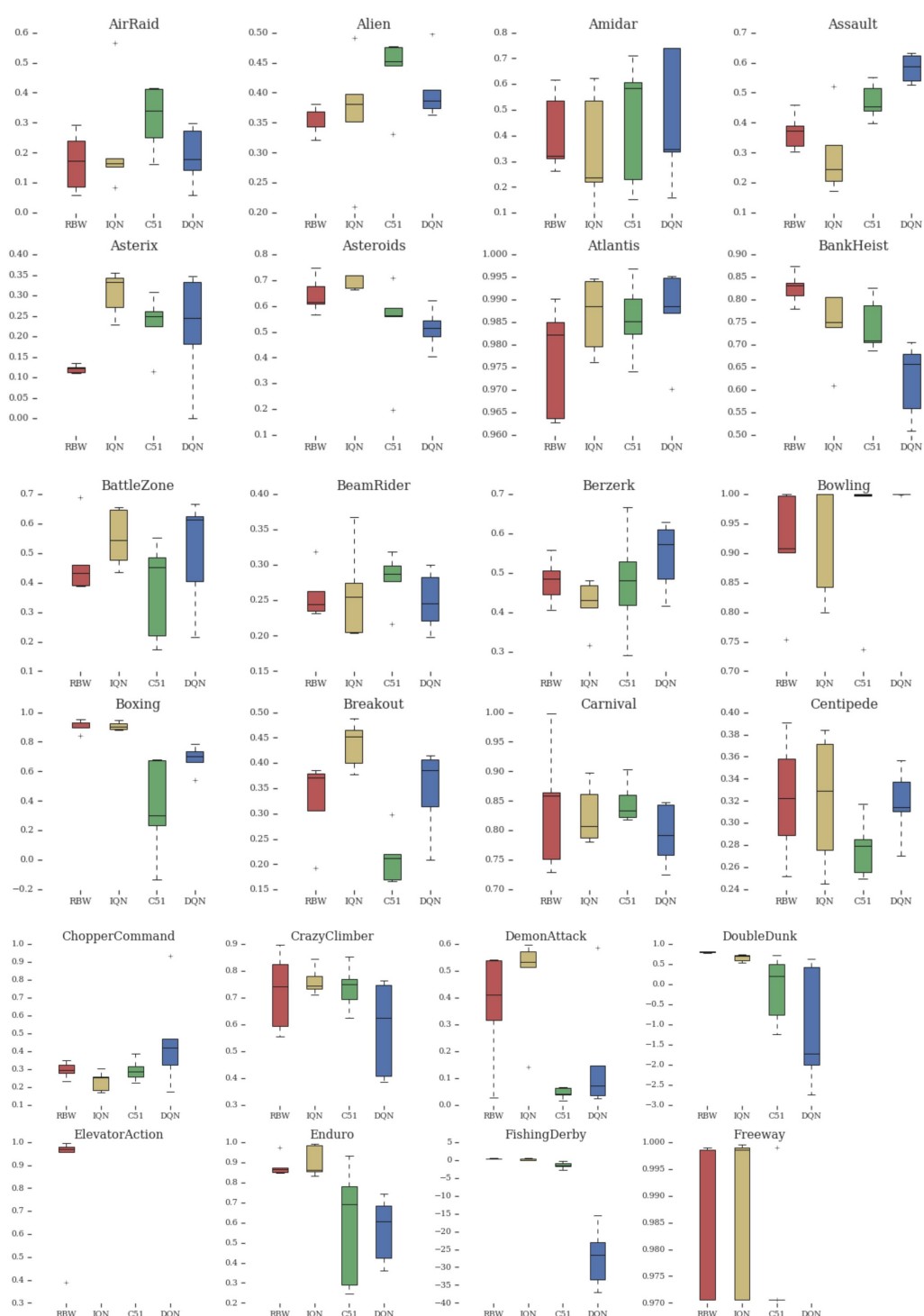

Figure 21: Risk across Fixed-policy Rollouts for DQN-variants tested on 60 Atari games, evaluated on a per-environment basis (page 1). Better reliability is indicated by more positive values. The x-axes indicate millions of Atari frames.

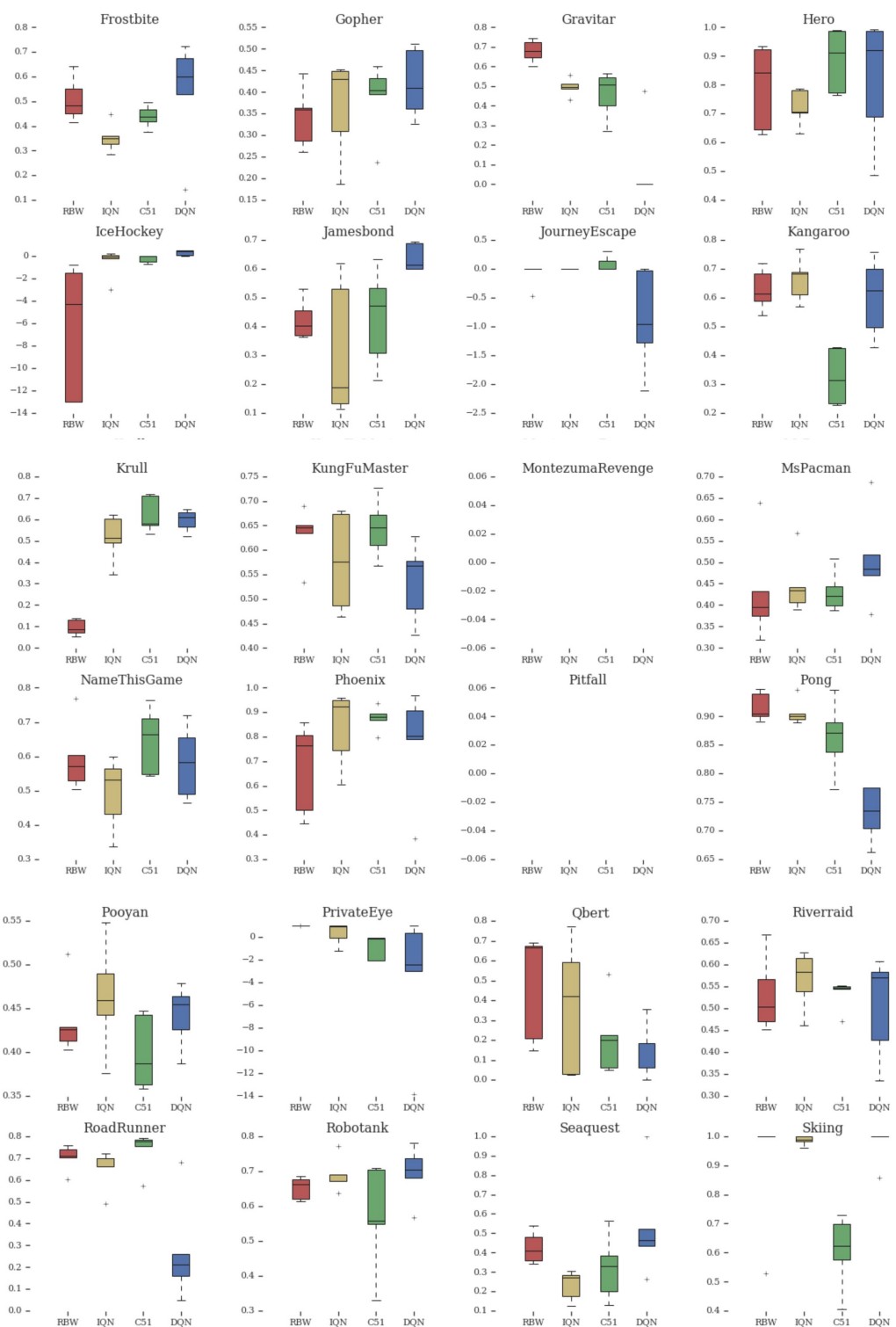

Figure 22: Risk across Fixed-policy Rollouts for DQN-variants tested on 60 Atari games, evaluated on a per-environment basis (page 2). Better reliability is indicated by more positive values. The x-axes indicate millions of Atari frames.

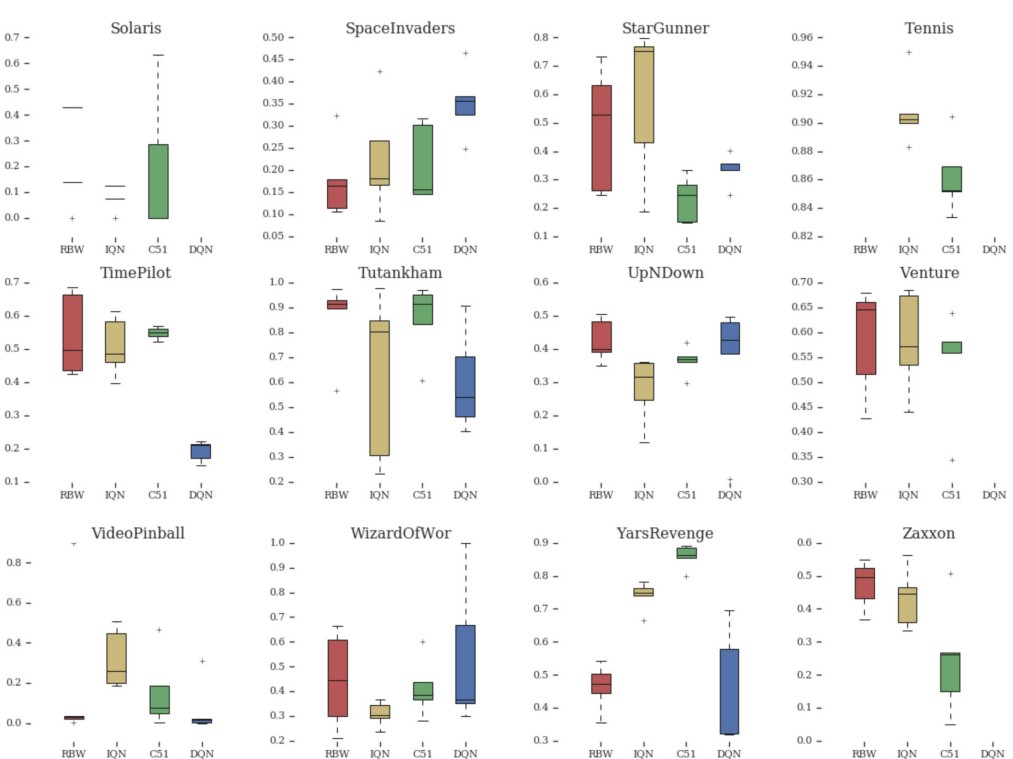

Figure 23: Risk across Fixed-policy Rollouts for DQN-variants tested on 60 Atari games, evaluated on a per-environment basis (page 3). Better reliability is indicated by more positive values. The x-axes indicate millions of Atari frames.

