# OpenReview forum: "Measuring the Reliability of Reinforcement Learning Algorithms"
_ICLR.cc/2020/Conference — Accept (Spotlight)_

### Official Review · AnonReviewer3 · 2019-10-16
**Official Blind Review #3**

**Rating:** 6

**Review:**

In this paper, the authors study an important problem in the area of reinforcement learning (RL). Specifically, the authors focus on how to evaluate the reliability of RL algorithms, in particular of the deep RL algorithms. The paper is well motivated by providing convincing justification of evaluating the RL algorithms properly. In particular, the authors define seven specific evaluation metrics, including 'Dispersion across Time (DT): IQR across Time', 'Short-term Risk across Time (SRT): CVaR on Differences', 'Long-term Risk across Time (LRT): CVaR on Drawdown', 'Dispersion across Runs (DR): IQR across Runs', 'Risk across Runs (RR): CVaR across Runs', 'Dispersion across Fixed-Policy Rollouts (DF): IQR across Rollouts' and 'Risk across Fixed-Policy Rollouts (RF): CVaR across Rollouts', from a two-dimension analysis shown in Table 1.

Moreover, the authors apply the proposed evaluation metrics to some typical RL algorithms and environments, and provide some insightful discussions and analysis.

Overall, the paper is well presented though it is somehow different from a typical technical paper.


**Experience Assessment:**

I do not know much about this area.

**Review Assessment: Checking Correctness Of Derivations And Theory:**

N/A

**Review Assessment: Checking Correctness Of Experiments:**

I assessed the sensibility of the experiments.

**Review Assessment: Thoroughness In Paper Reading:**

I read the paper at least twice and used my best judgement in assessing the paper.

---

> ### Author Response · Authors · 2019-11-13
> **Thank you for your positive comments**
>
> Thank you for your positive comments. We agree that this is an important area of study, and we hope that our work can have a beneficial effect on the field. As you have noted, we have taken a lot of care to construct these metrics and the surrounding procedures in a way that is rigorous and well motivated. Because this is a methods paper, we hold it as particularly important to present the methods in a digestible format, so we appreciate your recognition of this as well.

---

### Official Review · AnonReviewer1 · 2019-10-23
**Official Blind Review #1**

**Rating:** 8

**Review:**

*Summary*

Authors proposed a variety of metrics to measure the reliability of an RL algorithm. Mainly looking at Dispersion and Risk across time and runs while learning, and also in the evaluation phase.
Authors have further proposed ranking and also confidence intervals based on bootstrapped samples. They also compared the famous continuous control and discrete actions algorithms on Atari and OpenAI Gym on the metrics they defined.

*Decision*

I believe the paper is discussing a very important issue, and some possible solutions to it, even if not perfect it's an important step toward paying more attention to maybe similar metrics. I am in favor of the paper in general,  but I have some concerns.

1. My main concern is that why authors think that community will adopt these metrics and report them? I like how authors have proposed different metrics, but having one or two easy to compute metric is much more likely to be adopted, than 6 different metrics, which I’m not sure how easy it is to use the python package? It’s of main importance, because if community don’t use these metrics in the future, the contribution of the paper is minimal.

2. There is no question of the importance of reliability of RL algorithms, but we need to be careful that RL algorithms are not optimizing for metics like CVaR, so maybe a better learning algorithm (in the sense of expectation learning) might not have better reliability metrics because it is not the main objective.
So following this, how would authors think their metrics can be used to design a more reliable algorithms? For example there is good literature on CVaR learning for safe policies. Do you think there exists a proxy for metrics you introduced that can be used to for the objective of the optimization?

3. Another main concern is the effect of exploration strategy: All these metrics can be highly affected  by different exploration strategy in different environments. For example if an environment has a chain like structure, then given the exploration strategy you may have an extremely high CVaR or IQR. How do authors think they can strip off this effect? (Running all algorithm with the same exploration strategy is not sufficient, since the interplay of learning algorithm and exploration may be important)

4. Generalizability: How do authors think these metrics are generalizable. For example if algorithm A has better metrics than algorithm B on open AI Gym task for continuous control, how much we expect the same ranking applies while learning on a new environment. I am asking this, because to me, some of these metrics are very environment dependent, and being reliable in some environments may not imply reliability in other environments.


*Note*:
Code and modularity of it: The main contribution of this paper will be shown when other researchers start using it and report the metric, if the code is hard to use, the contribution of the paper is hard to be significant.

==== Post Rebuttal ====
Thanks for the responses authors posted, I think there is a good chance that the community will benefit from this experimental metrics in the future, so I increase my rating to accept.


**Experience Assessment:**

I have read many papers in this area.

**Review Assessment: Checking Correctness Of Derivations And Theory:**

N/A

**Review Assessment: Checking Correctness Of Experiments:**

I assessed the sensibility of the experiments.

**Review Assessment: Thoroughness In Paper Reading:**

N/A

---

> ### Author Response · Authors · 2019-11-13
> **Thank you for your insightful comments [part 1]**
>
> > I believe the paper is discussing a very important issue, and some possible solutions to it, even if not perfect it's an important step toward paying more attention to maybe similar metrics. I am in favor of the paper in general,  but I have some concerns.
>
> Thank you for recognizing the importance of this work. Thank you also for your insightful questions and comments. We respond to each comment individually below.
>
> > 1. My main concern is that why authors think that community will adopt these metrics and report them? I like how authors have proposed different metrics, but having one or two easy to compute metric is much more likely to be adopted, than 6 different metrics, which I’m not sure how easy it is to use the python package? It’s of main importance, because if community don’t use these metrics in the future, the contribution of the paper is minimal.
>
> We agree that this is an important question. An empirical answer is that we have already received strong interest in these metrics from RL researchers and engineers who are developing RL for real-world practical application. Reliability is a high priority for them, and these metrics (and the accompanying framework, including statistical tests and confidence intervals) fill a need that was previously unmet, to provide rigorous and quantitative measurement of reliability on different dimensions. We are already starting to work with these teams on integrating the metrics into daily tests etc.
>
> In previous presentations of this work, we have also received strong interest and enthusiasm from RL researchers. We believe that, sociologically, pure researchers will be incentivized to adopt these metrics too. As the field moves towards developing more reliable algorithms, as is already happening, researchers will look for ways to measure their gains in reliability. For example, Haarnoja et al would have benefited from using these metrics to demonstrate the reliability of SAC, especially given the tools to rigorously compare against other algorithms with statistical tests. Reviewers may also ask for evaluation of these metrics, given the growing awareness of RL reliability.
>
> Furthermore, we have taken great care to design our package to be as easy to use as possible, and we expect that this will greatly ease adoption. Installation will be straightforward as a Python package. The entire pipeline (evaluation of the metrics, computation of statistical tests and confidence intervals, and generation of plots) will be easy to run with just a few commands. The small number of parameters minimizes the cognitive load on users. The package will be compatible with a number of common data formats, including both Tensorflow and PyTorch outputs. These metrics will be well-integrated with a popular RL framework, which will  also encourage adoption. It will also be compatible with a number of other popular RL libraries. The package will also be open source so that users can inspect the code and easily adapt it to new use cases.
>
> We agree that having multiple metrics adds complexity to our framework; this is in fact something we have internally discussed. Ultimately, however, we believe that it is important to include these different metrics for because they measure reliability for different use cases. Upon release of the package, we plan to include documentation that provides clear examples of different usages for different scenarios.
>
> > 2. There is no question of the importance of reliability of RL algorithms, but we need to be careful that RL algorithms are not optimizing for metics like CVaR, so maybe a better learning algorithm (in the sense of expectation learning) might not have better reliability metrics because it is not the main objective.  So following this, how would authors think their metrics can be used to design a more reliable algorithms? For example there is good literature on CVaR learning for safe policies. Do you think there exists a proxy for metrics you introduced that can be used to for the objective of the optimization?
>
> We believe that it is indeed possible and a valuable idea to adopt some version of these metrics into the optimization function, analogous to prior work incorporating risk measures on cumulative returns. This would be a very interesting avenue for future work, and we would definitely be interested in investigating further.
>
> In our analysis, we evaluate the metrics on algorithms that were optimized for mean performance. As you pointed out, there may be variants of these algorithms that perform better on reliability, given a different objective function. We use this method of optimization because this is what practitioners typically use, and we wanted to present an analysis of algorithms as they are typically used. However, we certainly hope that this work will motivate researchers to inspect such metrics while tuning hyper parameters and designing objective functions.

---

> > ### Author Response · Authors · 2019-11-13
> > **Thank you for your insightful comments [part 2]**
> >
> > > 3. Another main concern is the effect of exploration strategy: All these metrics can be highly affected  by different exploration strategy in different environments. For example if an environment has a chain like structure, then given the exploration strategy you may have an extremely high CVaR or IQR. How do authors think they can strip off this effect? (Running all algorithm with the same exploration strategy is not sufficient, since the interplay of learning algorithm and exploration may be important)
> >
> > It is definitely true that exploration strategies affect reliability. In our analysis, the exploration strategies were fixed according to the method used in the original papers (the continuous control experiments) or in the Dopamine package release (the discrete control experiments). For many algorithms, it is difficult to separate out the exploration strategy from the algorithm itself. However, one could certainly imagine a scenario in which a user lets the governing hyperparameters (e.g. action selection noise or greediness) be free parameters are optimized. In this case, we expect that evaluation should be performed on the optimized algorithm.
> >
> > > 4. Generalizability: How do authors think these metrics are generalizable. For example if algorithm A has better metrics than algorithm B on open AI Gym task for continuous control, how much we expect the same ranking applies while learning on a new environment. I am asking this, because to me, some of these metrics are very environment dependent, and being reliable in some environments may not imply reliability in other environments.
> >
> > The values of these metrics are definitely environment dependent. As far as we understand, this is an inherent part of evaluating RL algorithms, whether on performance or reliability, because RL behavior depends on the specifics of the environment. For practical use cases, we expect users to evaluate on the environments in which they plan to deploy their algorithms. For researchers evaluating new algorithms, we hope that they evaluate on a range of environments (both for performance and reliability); this provides a fuller picture of an algorithm’s reliability, and also makes extrapolation to novel tasks more justifiable. Motivated by your and Reviewer 1’s comments, we have added per-environment evaluations of the metrics in Appendix F. Interestingly, the ordering of algorithms is actually relatively consistent across environments, though this is certainly not always the case.
> >
> > > Code and modularity of it: The main contribution of this paper will be shown when other researchers start using it and report the metric, if the code is hard to use, the contribution of the paper is hard to be significant.
> >
> > We strongly agree that ease of use is a critical component of this project, in addition to the more theoretical concerns that have been addressed in the paper. To this end, we have taken many steps to encourage adoption and to ensure ease of use (described above in our answer to Question 1). Please let us know if we can clarify further.

---

### Official Review · AnonReviewer2 · 2019-10-23
**Official Blind Review #2**

**Rating:** 8

**Review:**

This paper provides a unified way to provide robust statistics in evaluating RL algorithms in experimental research. Though I don't believe the metrics are particularly novel, I believe this work would be useful to the broader community and was evaluated on a number of environments. I do have a few concerns, however, about experimental performance per environment being omitted from both the main paper and the appendix.

Comments:

+ I think this is a valuable work and the ideas/metrics are useful, though I'm not sure I would call them novel (CVar and the like have been seen before).  I think the value comes in the unification of the metrics to give more robust pictures of algorithmic performance.
+ The details of all of these evaluations and individual performance should be provided in the appendix, however, it seems only MuJoco curves were included. Moreover, it says that a blackbox optimizer was used to find hyperparameters, but these hyperparameters were not provided in the appendix or anywhere else as far as I can tell. I think it's important for a paper which recommends evaluation methodology in particular to be more explicit regarding all details within the appendix. I hope to see additional details in future revisions -- including per-environment performance.
+ I believe clustering results across environments can be potentially misleading. Say that we have an environment where the algorithm always fails but is very consistent and an environment where it excels. These are blended together in the current Figures. While it requires more space, I believe it is important to separate these two. I am concerned that a recommendation paper like this one will set a precedent for only including the combined metrics of algorithmic performance across environments, masking effects.  I would suggest splitting out results per environment as well and pointing out particular cross-environment phenomena.

There is a missing discussion of prior work on statistical testing of RL evaluation:
+ Colas, Cédric, Olivier Sigaud, and Pierre-Yves Oudeyer. "A Hitchhiker's Guide to Statistical Comparisons of Reinforcement Learning Algorithms." arXiv preprint arXiv:1904.06979 (2019).
+ Colas, Cédric, Olivier Sigaud, and Pierre-Yves Oudeyer. "How many random seeds? statistical power analysis in deep reinforcement learning experiments." arXiv preprint arXiv:1806.08295 (2018).

EDIT: Score boosted after significant updates to the paper.


**Experience Assessment:**

I have published in this field for several years.

**Review Assessment: Checking Correctness Of Derivations And Theory:**

I assessed the sensibility of the derivations and theory.

**Review Assessment: Checking Correctness Of Experiments:**

I carefully checked the experiments.

**Review Assessment: Thoroughness In Paper Reading:**

I read the paper at least twice and used my best judgement in assessing the paper.

---

> ### Author Response · Authors · 2019-11-13
> **Thank you for your comments and your thoughtful consideration of our work**
>
> > This paper provides a unified way to provide robust statistics in evaluating RL algorithms in experimental research. Though I don't believe the metrics are particularly novel, I believe this work would be useful to the broader community and was evaluated on a number of environments. I do have a few concerns, however, about experimental performance per environment being omitted from both the main paper and the appendix.
>
> Thank you for your comments and for your recognition of the value of this work. We appreciate your thoughtful consideration of our paper, and we respond below to your suggestions in detail.
>
> > I think this is a valuable work and the ideas/metrics are useful, though I'm not sure I would call them novel (CVar and the like have been seen before).  I think the value comes in the unification of the metrics to give more robust pictures of algorithmic performance.
>
> Thank you for your recognition of the usefulness of these metrics. We agree that developing a unified framework for measuring reliability is of value to the community, and this has been a strong motivator for us in doing this work. With regard to CVaR in RL, our understanding is that it has previously been applied to the cumulative returns within an episode, but we are not aware that it has been applied to the variables pointed to in the paper, which measure distinct aspects of risk and reliability. We believe that our framework additionally introduces explicit delineations of different dimensions of reliability, definitions of quantitative metrics for measuring on these dimensions, best practices for pre- and post-processing, and rigorous statistical tests and confidence intervals that allow aggregation across tasks while respecting the measurements as being repeated measures on different algorithms and tasks. Notwithstanding, we agree that the word “novel” may be misunderstood, and we have removed it from our paper.
>
> > The details of all of these evaluations and individual performance should be provided in the appendix, however, it seems only MuJoco curves were included. Moreover, it says that a blackbox optimizer was used to find hyperparameters, but these hyperparameters were not provided in the appendix or anywhere else as far as I can tell. I think it's important for a paper which recommends evaluation methodology in particular to be more explicit regarding all details within the appendix. I hope to see additional details in future revisions -- including per-environment performance.
>
> We have added hyperparameters for both the continuous control and discrete control algorithms in Appendix E. This includes the search space used for the blackbox optimizer. For the Atari training curves, we previously had a pointer in Appendix D linking to the relevant part of the Dopamine project, but this was far too hidden so we have added a pointer in the main text instead. Please let us know if there are any other details that you believe we should include. We strongly agree that, given this is a methodology paper, we should hold ourselves to a high standard in this regard.
>
> We have also added per-environment metric results in Appendix F. Please see the next comment for more details.
>
> > I believe clustering results across environments can be potentially misleading. Say that we have an environment where the algorithm always fails but is very consistent and an environment where it excels. These are blended together in the current Figures. While it requires more space, I believe it is important to separate these two. I am concerned that a recommendation paper like this one will set a precedent for only including the combined metrics of algorithmic performance across environments, masking effects.  I would suggest splitting out results per environment as well and pointing out particular cross-environment phenomena.
>
> Thank you for making this important point. We have added per-environment results in Appendix F, and a pointer to those results in the main results in Section 5.5. We have also added the following text in Section 3 to explicitly encourage per-environment investigations as part of our recommendations: “Per-environment analysis -- The same algorithm can have different patterns of reliability for different environments. Therefore, we recommend inspecting reliability metrics on a per-environment basis, as well as aggregating across environments as described above.”
>
> > There is a missing discussion of prior work on statistical testing of RL evaluation:
> > [...]
>
> Thank you for pointing us to these papers, which are very relevant for RL practitioners who wish to be rigorous in their experiments and analysis. We have included a discussion of this work in the introduction.

---

> > ### Author Response · Authors · 2019-11-13
> > **Example analysis of cross-environment phenomena**
> >
> > To provide an example of a per-environment analysis, and to emphasize the importance of doing so, we have also added the following text to the results in Section 5.5:
> >
> > "To see metric results evaluated on a per-environment basis, please refer to Appendix F. Rank order of algorithms was often relatively consistent across the different environments evaluated. However, different environments did display different patterns across algorithms. For example, even though SAC showed the same or better Dispersion across Runs for most of the MuJoCo environments evaluated, it did show slightly worse Dispersion across Runs for the HalfCheetah environment (Fig 7a). This kind of result emphasizes the importance of inspecting reliability (and other performance metrics) on a per-environment basis, and also of evaluating reliability and performance on theenvironment of interest, if possible."

---

### Decision · Program_Chairs · 2019-12-19

**Decision:**

Accept (Spotlight)

**Comment:**

Main content:

This paper provides a unified way to provide robust statistics in evaluating the reliability of RL algorithms, especially deep RL algorithms. Though the metrics are not particularly novel, the investigation should be useful to the broader community as it compares seven specific evaluation metrics, including 'Dispersion across Time (DT): IQR across Time', 'Short-term Risk across Time (SRT): CVaR on Differences', 'Long-term Risk across Time (LRT): CVaR on Drawdown', 'Dispersion across Runs (DR): IQR across Runs', 'Risk across Runs (RR): CVaR across Runs', 'Dispersion across Fixed-Policy Rollouts (DF): IQR across Rollouts' and 'Risk across Fixed-Policy Rollouts (RF): CVaR across Rollouts'. The paper further proposed ranking and also confidence intervals based on bootstrapped samples, and compared against continuous control and discrete actions algorithms on Atari and OpenAI Gym.

--

Discussion:

The reviews clearly agree on accepting the paper, with a weak accept coming from a reviewer who does not know much about this subarea. Comments are mostly just directed at clarifications and completeness of description, which the authors have addressed.

--

Recommendation and justification:

This paper should be accepted due to its useful contributions toward doing a better job of measuring performance of RL.